# Reconstruction of a Segment of the UNESCO World Heritage Hadrian's Villa Tunnel Network by Integrated GPR, Magnetic–Paleomagnetic, and Electric Resistivity Prospections

**Annalisa Ghezzi [1],\*, Antonio Schettino [1], Pietro Paolo Pierantoni [1], Lawrence Conyers [2], Luca Tassi [1], Luigi Vigliotti [3], Erwin Schettino [4], Milena Melfi [5], Maria Elena Gorrini [6] and Paolo Boila [7]**

[1] School of Science and Technology, University of Camerino, 62032 Camerino, Italy
[2] Department of Anthropology, University of Denver, Denver, CO 80210, USA
[3] Istituto di Scienze Marine, CNR, 40129 Bologna, Italy
[4] Instituto Andaluz de Ciencias de la Tierra, CSIC—Universidad de Granada, 18100 Granada, Spain
[5] Faculty of Classics, University of Oxford, Oxford OX1 3LU, UK
[6] Dipartimento di Studi Umanistici, Università degli studi di Pavia, 27100 Pavia, Italy
[7] Idrogeotec S.N.C., 06127 Perugia, Italy
\* Correspondence: annalisa.ghezzi@unicam.it; Tel.: +39-0737-402641

**Abstract:** Hadrian's Villa is an ancient Roman archaeological site built over an ignimbritic tuff and characterized by abundant iron oxides, strong remnant magnetization, and elevated magnetic susceptibility. These properties account for the high-amplitude magnetic anomalies observed in this site and were used as a primary tool to detect deep archaeological features consisting of air-filled and soil-filled cavities of the tuff. An integrated magnetic, paleomagnetic, radar, and electric resistivity survey was performed in the Plutonium-Inferi sector of Hadrian's Villa to outline a segment of the underground system of tunnels that link different zones of the villa. A preliminary paleomagnetic analysis of the bedrock unit and a high-resolution topographic survey by aerial photogrammetry allowed us to perform a computer-assisted modelling of the observed magnetic anomalies, with respect to the archaeological sources. The intrinsic ambiguity of this procedure was reduced through the analysis of ground penetrating radar and electric resistivity profiles, while a comprehensive picture of the buried archaeological features was built by integration of the magnetization model with radar amplitude maps. The final subsurface model of the Plutonium-Inferi complex shows that the observed anomalies are mostly due to the presence of tunnels, skylights, and a system of ditches excavated in the tuff.

**Keywords:** archaeological geophysics; magnetic methods; ground penetrating radar; tunnel detection; data integration

## 1. Introduction

### 1.1. Archaeological Background

Hadrian's Villa is a UNESCO World Heritage site near Rome (Figure 1). It was built, starting from 117–118 A.D., as one of the residences of the Roman Emperor Hadrian. Its site was chosen for its scenic location, proximity to Rome, and the presence of four aqueducts directed to Rome. The archaeological structures of this large villa include many monumental buildings and an important underground

network of tunnels that was created to link different parts of the complex and, possibly, for the transport of supplies [1]. Presently, only a small part of this network can be travelled by non-speleologists. The first systematic description of Hadrian's Villa was carried out by Pirro Ligorio in the 16th century [2]. Ligorio was the first to draw a complete map of the site and to attempt an identification of the function of any specific feature of the villa by assigning names to its various parts. One century later, Contini drew a more accurate map of the system of tunnels that run beneath the villa on the basis of the earlier study by Ligorio [3]. His map was accepted without substantial modifications by all of the following draughtsmen, including Piranesi ([1] and references therein). However, an important contribution to the characterization of the architectural elements of the villa was provided at the beginning of the 19th century by Reference [4], who accurately described and recorded both exposed structures and the accessible subterranean features (already mapped by earlier authors).

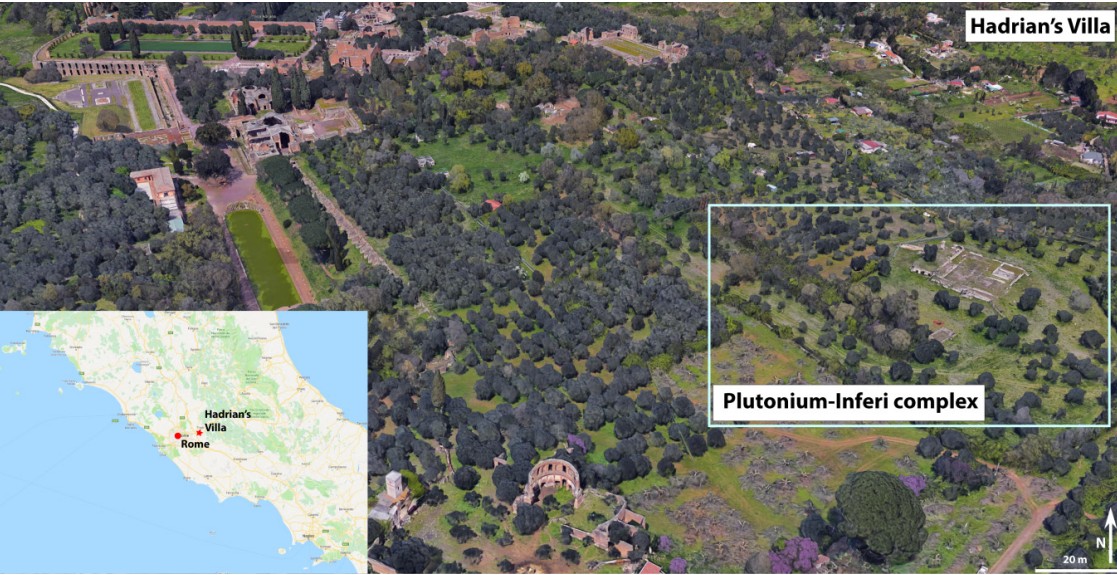

**Figure 1.** Hadrian's Villa archaeological site.

More recently, an accurate topographic survey of the network of tunnels was performed by Reference [1] (Figure 2), who divided the underground features in four categories according to their functionality. The first category includes brick dressed cryptoportica and ambulacra. The second category comprises underground tunnels for use of both pedestrians and carts, occasionally interrupted by open-air segments. The third category includes underground pedestrian passages linking different buildings of the villa, smaller in size than those of the former category. Finally, the last category consists of a heterogeneous set of basements and underground rooms with different purposes.

The buildings in the Inferi-Plutonium area have been generally understood as related to the belief of an after-life existence and the cult of death [5]. According to this hypothesis, which dates back to the first explorations by Pirro Ligorio, the Inferi, a large ditch dug in the tuff, would represent the River Styx and the entrance to the Underworld, while the Plutonium would be a temple devoted to Pluto, king of the Underworld.

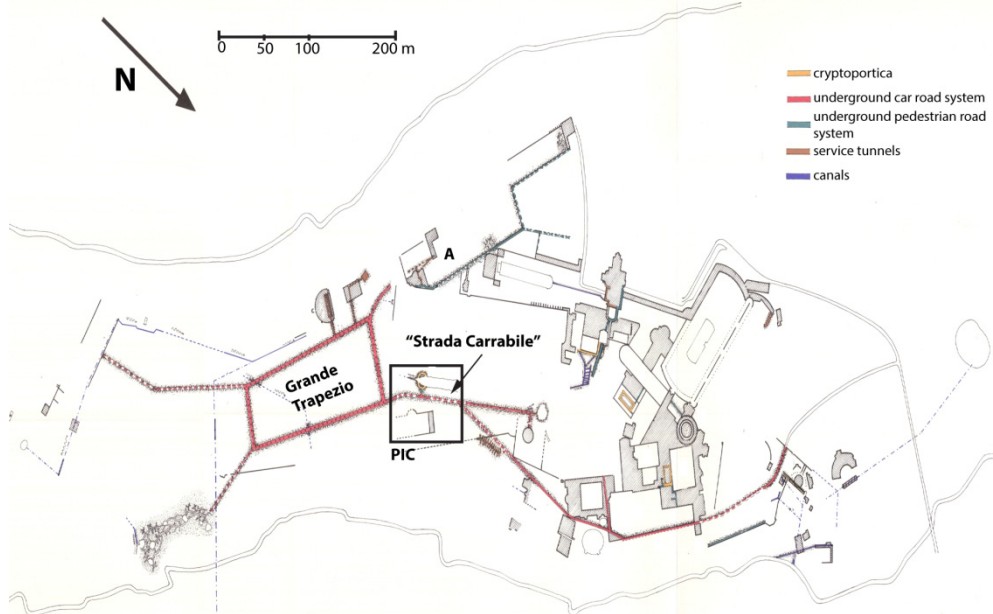

**Figure 2.** Hadrian's Villa general plan (modified from Reference [1]). **PIC** = Plutonium-Inferi complex; **A** = Accademia.

### 1.2. Geological Setting

Hadrian's Villa lies over the Quaternary Colli Albani volcanic district, a ~600 ka to present undersaturated K-rich magmatic province that is part of the ~250 km long peri-Thyrrhenian volcanic belt [6]. The geological framework of this site consists of a complex volcano-sedimentary succession made of ignimbrite sheets interposed with fall deposits and lava flows [7]. The irregularities in geometry and thickness, as well as lateral facies variations and heterogeneous alteration of the stratigraphic units, are related to the paleotopographic control on the transport and depositional mechanisms of the volcanic products. The outcropping lithology at Hadrian's Villa consists of the Pozzolanelle ignimbrite, the upper depositional unit of the Villa Senni eruption unit, which represents the youngest (355 ka) mafic caldera-forming eruption of the Colli Albani [8,9]. The Pozzolanelle unit is a chaotic ignimbritic tuff massive deposit, ranging in thickness from <2 to 40 m, and characterized by an ash matrix support texture abundant with phenocrystals (leucite, clinopyroxene, and biotite), holocrystalline xenoliths, and reddish vesicular scoria clasts [10] (Figure 3).

The magnetic properties of the Colli Albani volcanic district have been extensively investigated over the last two decades e.g., [11]. These rocks have a relevant magnetic susceptibility, $\chi$, ranging between $4400 \times 10^{-6}$ and $32{,}400 \times 10^{-6}$ SI units, with an average value of $14{,}300 \times 10^{-6}$ SI units [11]. The natural remanent magnetization (NRM) of the Villa Senni unit is a single component thermoremanent magnetization (TRM) acquired in a short time interval during the cooling of the pyroclastic flow [12]. The magnetization intensity of the upper part of this unit varies between $14.7 \times 10^{-2}$ and $4.9$ A m$^{-1}$, while the mean paleomagnetic direction of the most stable component was found to be $D = 357.4°$, $I = 62.3°$ ($k = 23.6$, $\alpha_{95} = 3.5°$) [12].

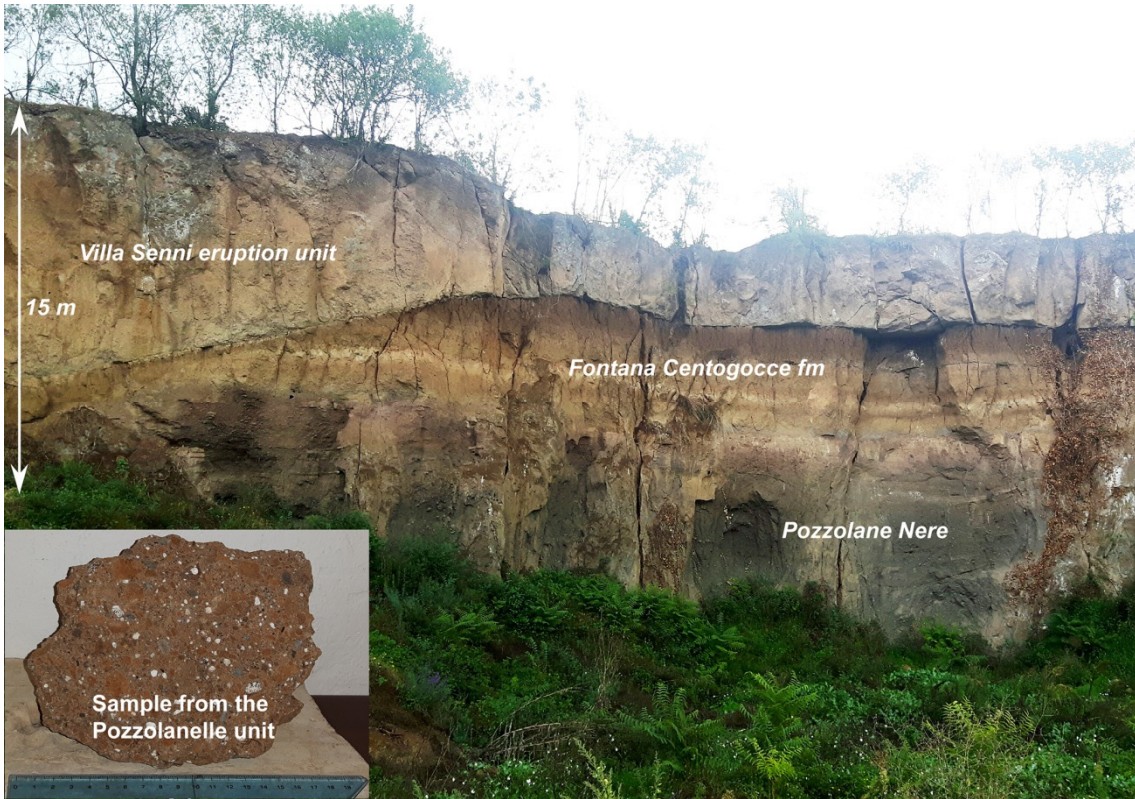

**Figure 3.** Stratigraphic section of the substratum of Hadrian's Villa. The picture was taken along the Via Tiburtina, close to the highway entrance to Tivoli.

### 1.3. Previous Geophysical Investigations at Hadrian's Villa

Geophysical methods have been widely used in archaeological research since at least the 1950s [13]. Despite the paramount archaeological importance of Hadrian's Villa, there is a very scarce record of published geophysical investigations at this site. Franceschini and Marras [14] applied electric resistivity methods for the first time to reconstruct the layout of the subterranean tunnels in the area of the Accademia (Figure 2). Their results substantially confirmed the map of Piranesi for this sector of the villa. A ground penetrating radar (GPR) survey of the area around the Plutonium-Inferi complex (Figure 1) was undertaken in 2016, by a team of the British School at Rome [15], to facilitate successive excavations in the context of a joint project of the universities of Oxford and Pavia [16]. The radar survey was performed using a 400 MHz antenna, which did not allow sufficient penetration for investigating the system of tunnels that had been dug in the tuff units underlying the Plutonium-Inferi area. The maximum penetration depth of this study was ~100 cm, thereby the resulting amplitude maps only showed archaeological features buried in the topsoil and not the cavities in the underlying rock formation. To date, the most used geophysical technique at Hadrian's Villa has probably been the laser scanner (an interesting example can be found in Reference [17]). Finally, the sophisticated surveying techniques (probes and robots) used in recent years by the Sotterranei di Roma, which explored a segment of the underground road system known as the "Strada Carrabile", are worth mentioning [18].

### 1.4. Integration of Geophysical Datasets

The application of integrated geophysical techniques in archaeological research is a widely used method that can produce a greater quantity of information than the individual datasets, because each technique may reinforce or validate what is observed by other approaches, besides to detect additional features [19]. All the geophysical methods provide, directly or after a modelling step, distributions

of "anomalies" of physical parameters in the underground. In this context, the word "anomaly" is widely used to indicate a strong localized contrast of some quantity (e.g., a reflection amplitude) with respect to the surrounding region, although this lexicon introduces some confusion with the classical potential field anomalies. The latter, in fact, are only remote expressions, observed near the Earth's surface, of real physical contrasts in the ground (e.g., differences of magnetization between soil and archaeological features). Integration of geophysical datasets is a practice that allows for the combination of distributions of different physical variables at some depth in the subsurface in order to obtain a comprehensive picture of the pattern of buried archaeological features.

There are three broad categories of processing methods for the combination of raster data in archaeological geophysics, as follows [20,21], and references therein: (1) Computer graphics methods, (2) mathematical transformations, and (3) statistical overlays. A common computer graphics method used to combine two or three scalar fields is based on the simultaneous assignment of the corresponding normalized data sets to the R, G, and B channels of a color map. All the methods based on grid mathematics, either logic or algebraic, for example Boolean unions, linear combinations, etc., e.g., [22], fall in the second category. A major problem with these two approaches is given by the fact that the different physical quantities used to map the same archaeological features generate, in most cases, structural lineaments that are slightly displaced, with respect to each other, due to measurement errors, instrument sensitivity, etc. Consequently, these maps may display very low correlation, even when a buried object is characterized by significant contrasts of such different physical quantities [23]. To overcome this problem, it is possible to apply statistical methods of data integration, for example local Pearson [23] and principal component analysis [24]. In any case, it is important to note that none of these techniques should be used to combine magnetic anomalies or vertical gradients of the total magnetic field intensity with grids that represent the distribution of a physical quantity in the ground. The reason is that magnetic anomalies and vertical gradients observed at any location close to the Earth's surface are related in a very complex way to all the existing sources in the underground, at any depth, and with a lateral extent of several tens of meters, not only to sources just below the observation point.

However, in a broader context, geophysical data integration may involve additional techniques, beyond the set of mathematical, statistical, or logical operations that can performed to combine data grids and produce images for the visual interpretation and localization of archaeological structures. In reality, in the acceptation of Harris et al. [25], the class of methods mentioned above is referred to as *data combination* (or *data fusion*, *data merging*), while the word *integration* should be used only when a modelling of the data is used to produce a thematic map. In this paper, we present a new approach to the integration of magnetic and GPR data, based on the preliminary conversion of observed magnetic anomalies into a distribution of magnetization that potentially describes the correct shape, size, location, and, possibly, the material of the buried archaeological features. This conversion is accomplished through a computer-assisted forward modelling procedure, constrained by the analysis of GPR and electric resistivity tomography (ERT) profiles. In fact, the construction of a magnetization model starting from the observed magnetic anomalies is affected by ambiguity (i.e., non-uniqueness of the magnetization distribution), due both to intrinsic characteristics of potential field geophysics and to the presence of uncertainty in the total field observations [26,27]. In our approach, the independent observation of archaeological features on GPR and ERT profiles allows for constraining the magnetization model and eliminating alternative solutions that would generate equivalent anomalies. The final outcome consists into a more reliable magnetization model of the ground, which provides a physical representation of the buried structures. Differently from the observed magnetic anomalies, such magnetization distribution can be effectively combined with GPR amplitude slices to generate a comprehensive thematic magnetization map representative of all the buried archaeological features, which also includes objects that are not detected by one of the two methods. In this paper, we describe the application of integrated geophysical methods to one of the

least known sectors of Hadrian's Villa, the Plutonium-Inferi complex (Figure 1), with the objective of reconstructing a segment of the underground tunnel network in this area (Figure 4).

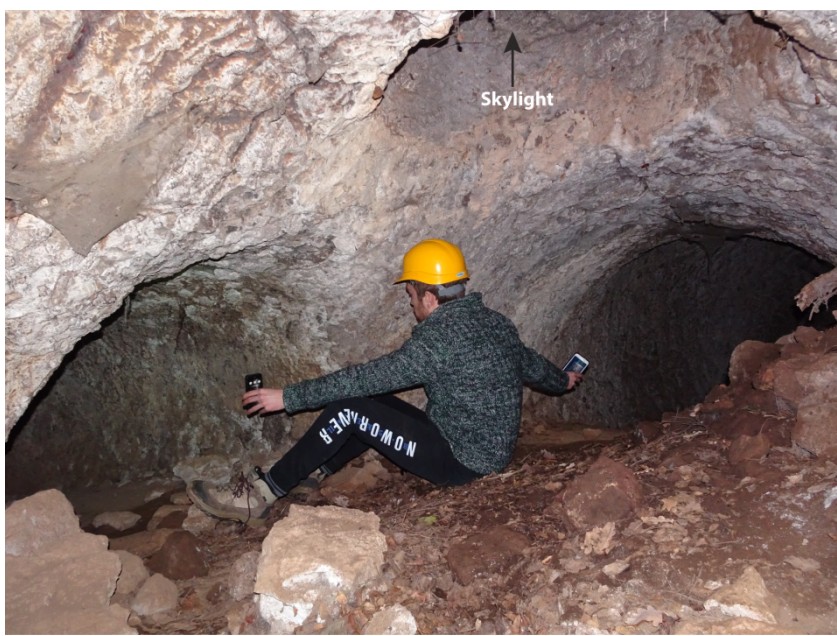

**Figure 4.** Tunnels in the Plutonium-Inferi complex.

### 1.5. Rationale for Magnetic Field Modelling in the Plutonium-Inferi Area

The present study aims at detecting cavities in the tuff units by combining several sources of data, obtained from different geophysical methods, as follows: Electric resistivity, magnetometry, GPR, paleomagnetism, and aerial photogrammetry. The strong magnetic susceptibility, $\chi$, and NRM of the tuff units on which Hadrian's Villa was built has amplified the contrast with the buried structures and the cavities in the substratum, so that the corresponding magnetic anomalies reach amplitudes up to ~2200 nT. In a sense, the bedrock geology of this site is a unique example of a natural "amplifier" of magnetic anomalies associated with archaeological features. Consequently, we used the magnetic data set as the primary source for reconstructing the layout of the tunnel network by using a technique of forward modelling, but the whole procedure and the results were constrained and assessed, respectively, by the other sources of data.

Our approach to collecting, processing, and modelling magnetic data at archaeological sites has been described in previous works [26,27]. Such a procedure requires a precise digital elevation model (DEM) of the survey area and a knowledge of the magnetization parameters of the ground. In most cases, a measure of the average magnetic susceptibility of the soil is sufficient to set up a magnetization model, but in special conditions, like those encountered at Hadrian's Villa, it is also necessary to know the NRM and the susceptibility of the bedrock formation just below the soil layer, because some of the structures were dug directly in the tuff. In this study, a DEM for the Plutonium-Inferi area was created using aerial photogrammetry techniques, while the magnetic parameters of the ground were obtained through a paleomagnetic study of the Pozzolanelle unit and soil susceptibility measurements. Although a mean paleomagnetic direction and NRM was available for the whole Villa Senni unit [12], the variability of magnetization intensity suggested an independent determination of these parameters for the Hadrian's Villa area. In the next sections, we first discuss the various techniques used for the data acquisition and processing, then we show how the different sources of data were integrated in order to reconstruct the topology of the tunnel network underneath the Plutonium-Inferi area. Our results confirm the presence of tunnels, whose existence was already known, but also reveal the existence of additional underground tunnels and of a system of ditches dug at the top of the tuff, which probably served as irrigation channels.

## 2. Methods

In this section, we describe in detail the acquisition and processing methods for each of the geophysical techniques used in this survey.

### 2.1. GPR Data Acquisition Procedures

GPR data were acquired using a GSSI SIR 4000 system equipped with a 200 MHz antenna, which, in principle, should have allowed sufficient penetration for investigating the system of tunnels that had been dug in the tuff units underlying the Plutonium-Inferi complex. However, the soil of this area includes abundant clays that formed by weathering of the volcanic substratum. In moist conditions, these clays display strong electric conductivity and attenuation, thereby the maximum depth of penetration did not generally exceed 2.5 m. GPR data were acquired using the following basic parameters:

- Survey mode  =  Distance mode (with odometer);
- Scans/m       =  50;
- Samples/scan  =  1024;
- Range         =  300 ns;
- Bits/sample   =  32;
- Line spacing  =  0.5 m;
- Antenna       =  GSSI 5106, 200 MHz, shielded antenna

To improve the coupling with the ground, the survey lines were travelled using three operators, one at the console and two who drove the large 5106 antennae along the track lines (Figure 5). Other people were responsible for the setup of the survey geometry, for the GPS positioning, and for the deployment of metric tapes. A local reference frame was established in order to combine the GPR data coming from different areas. We divided the survey region into 13 areas (Figure 6) of variable geometry.

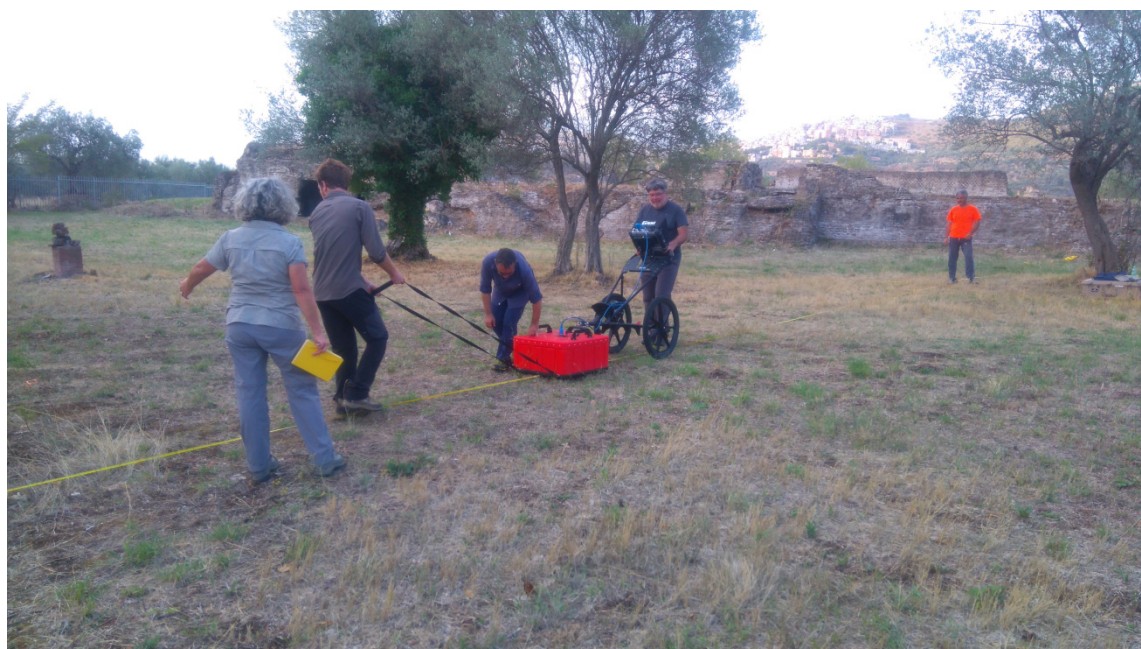

**Figure 5.** GPR system used to investigate the area and arrangement of operators.

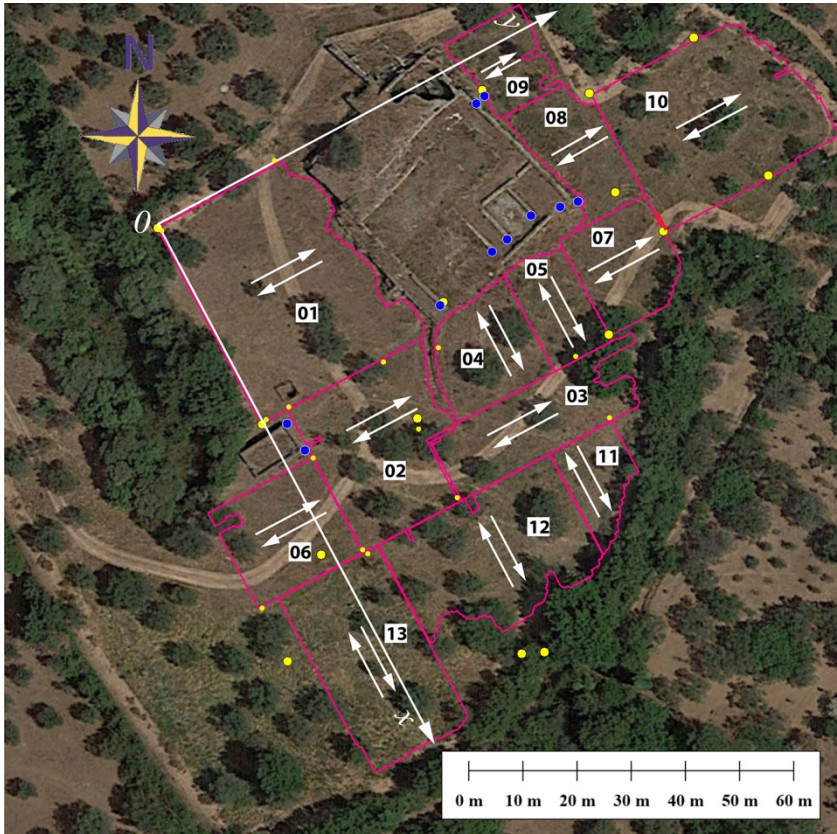

**Figure 6.** System of 25 GPS points used to define the local reference frame (yellow and blue dots) and the GPR survey areas (red lines). The direction of the survey lines is indicated by the white arrows. The coordinate axes (*x*,*y*) show the best-fitting reference frame through the assigned GPS points. The blue dots indicate locations used as control points for aerial photogrammetry.

In all cases, the survey lines were travelled in zig-zag mode (one direction) and the local coordinates of the end points were recorded by an operator directly in the field. A set of 25 reference points was established with assigned local coordinates ($_i$, $_i$) and known geographic coordinates ($x_i$, $y_i$), obtained by GPS measurements performed by a Leica 1200 system, with 1 cm accuracy, and expressed in the UTM 33 reference frame. A computer algorithm then calculated the best-fitting rigid transformation from local coordinates to UTM by weighted least squares minimization of corner location errors, relative to the measured GPS locations. The best-fitting orientation of the local reference frame relative to the UTM datum is illustrated in Figure 6, along with the GPS reference points. The misfit between the reference point locations and the best-fitting UTM coordinates resulted to be less than 0.56 m.

## 2.2. GPR Data Processing

Processing of GPR data was performed using the software listed in Table 1. The data underwent the following basic processing steps:

- Automatic gain control;
- Time-zero correction;
- Band pass filtering (~100–300 MHz);
- Background removal;
- Kirchhoff migration;
- Creation of amplitude slices;
- Normalization;
- Equalization; and

- Knitting

**Table 1.** Software used in data processing.

| Software | Author/Company | Usage |
|---|---|---|
| GPR Viewer 1.8.5 | J. E. Lucius & L. B. Conyers | Analysis of GPR profiles |
| GPR Slice v7 | Geophysical Archaeometry Laboratory | Basic processing of GPR data and generation of amplitude slices |
| Magmap 2000 v. 5.04 | Geometrics | Magnetic data pre-processing |
| Oasis Montaj 8.3 | Geosoft | Magnetic and GPR amplitude grid processing |
| Res2Dinv 3.54 | Geotomo Software | Electric resistivity inversion |
| Photoscan 1.4.5 | Agisoft | Aerial photogrammetry |
| Thopos 2018 | Studio Tecnico Guerra | Aerial photogrammetry |
| MatLab R2017 | MathWorks | Polynomial regression of magnetic field intensity data |
| ArchaeoMag 2.5 | A. Schettino & A. Ghezzi | Forward modelling of magnetic data |

To perform the data migration, we searched and fit reliable hyperbolae for each area. Then, laterally homogeneous velocity models were built, subdividing the substratum into horizontal layers, different for each area. The average velocity, $<v>$, and relative dielectric permittivity, $<\varepsilon>$, of each area of Figure 6 are listed in Table 2. These parameters were calculated over a reference two-way travel time (TWTT) interval, $T = 100$ ns, starting from the pairs $(T_k, v_k)$ provided by the velocity analysis, $T_k$ and $v_k$ being the TWTT to the bottom of layer $k$ and the corresponding velocity through the same layer.

**Table 2.** Average velocity (in cm ns$^{-1}$) and dielectric constant of the 13 areas of Figure 6.

| | 01 | 02 | 03 | 04 | 05 | 06 | 07 | 08 | 09 | 10 | 11 | 12 | 13 |
|---|---|---|---|---|---|---|---|---|---|---|---|---|---|
| $<v>$ | 6.7 | 9.3 | 6.6 | 5.6 | 8.2 | 5.0 | 7.3 | 4.4 | 6.3 | 6.5 | 7.0 | 6.7 | 7.0 |
| $<\varepsilon>$ | 20.1 | 10.5 | 20.7 | 28.5 | 13.4 | 35.7 | 17.1 | 45.8 | 22.4 | 21.5 | 18.3 | 19.8 | 18.4 |

They are given, respectively, by the following:

$$\langle v \rangle = \frac{1}{T} \sum_k v_k (T_k - T_{k-1}) \tag{1}$$

$$\langle \varepsilon \rangle = \frac{c^2}{\langle v \rangle^2} \tag{2}$$

where $c = 30$ cm ns$^{-1}$ is the speed of light.

The GPR surveys were performed in a time interval of two years and in different environmental conditions. Consequently, strong lateral variations in average dielectric permittivity resulted for the 13 areas (see Table 2), depending, in large part, from the variable water content of the soil and the underlying tuff. At the next step, ~30 cm thick amplitude slices were generated for all the 13 areas, with a 50% overlap between successive slices. Such thickness was a compromise value selected, taking the minimum range resolution resulting from velocity analysis, $\delta z \geq v_{\max}/(4f_c) = 17.4$ cm ($f_c$ = central frequency), and the necessity of enhancing structures with significant thickness into account, more relevant for the objectives of this research.

From this data set, we chose four meaningful time slices that were representative of shallow, intermediate, and deep burial depths. These slice grids were assigned local coordinates in the survey reference frame (Figure 6), according to their relative position in the mosaic of areas, and normalized

to reduce amplitude differences associated with diverse conditions of the ground, as well as with the different gains applied to the raw data. In general, the range of amplitudes recorded in slices of any depth interval for the various areas that compose a survey region are very different each other, depending on the environmental conditions during the acquisition step and from the applied gains. Consequently, the direct construction of a composite mosaic from these grids may produce poor results, because colors in the final map are assigned on the basis of a statistical classification performed on the whole mosaic (histogram equalization). To overcome this problem, it is possible to perform georeferencing and display independently for each area and compose only the final images in a GIS project file. In the alternative approach, followed here, the individual grids of each area were composed in local coordinates to create a unique mosaic for the whole survey region. The advantage is that a unique grid of reflection amplitudes for the study area is available for further processing, data integration, etc. In this instance, it is necessary to normalize all the component grids before their knitting, but the quality of the final mosaic can be still improved, ensuring continuity along the edges of each component grid and eliminating small reflections that are not generally associated with archaeological features. To this purpose, we applied an equalization procedure that isolated the relevant information from each grid and, at the same time, allowed a coherent representation of structures that crossed area boundaries. In this procedure, the final composite grid was built iteratively by the addition of successive component grids. At each iteration step, before its inclusion in the mosaic, each normalized grid $G$ underwent the following transformation:

$$G'(x,y) = \begin{cases} \alpha G(x,y) \text{ if } G(x,y) \geq M \\ 0 \text{ otherwise} \end{cases} \tag{3}$$

where $\alpha$ is a scaling factor and $M < 1$ is simply a threshold that is selected after visual inspection to eliminate or reduce background noise not related to real archaeological structures. The choice of the scaling factor $\alpha$ was performed in such a way that structures crossing the boundary between the actual mosaic and the new grid had approximately the same amplitude.

### 2.3. Magnetic Field Intensity Acquisition

Total field magnetic intensities were collected using a Geometrics G-858 caesium vapor magnetometer. The whole survey area was divided in 10 rectangles having variable dimensions, which were assembled at the end of the processing in the UTM 33N reference frame (Figure 7).

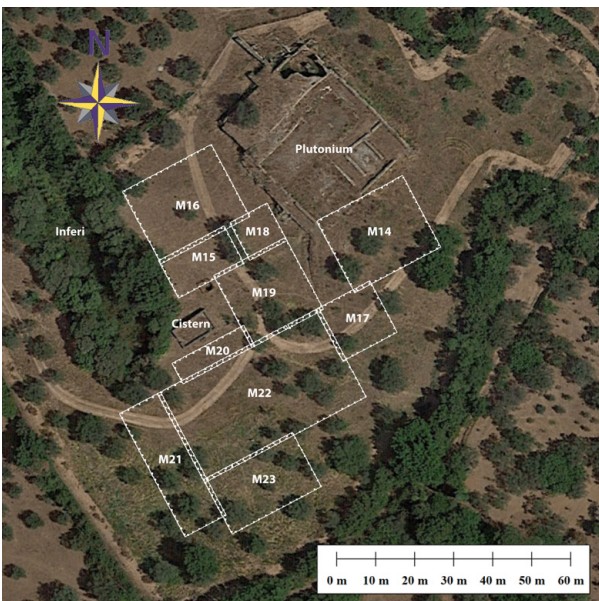

**Figure 7.** Area covered by the magnetic survey (white boxes).

The magnetic data were acquired at 10 Hz frequency (corresponding to an average 11 cm distance between successive readings) along survey lines equally spaced at 0.5 m and in one direction, using 5 m spaced markers to reduce navigation errors. All the total field measurements were performed in solar-quiet conditions, with the *Kp* index not exceeding 2, and the data underwent standard pre-processing, consisting of despiking and decorrugation procedures.

## 2.4. Magnetic Field Processing

The total field data of each area were decorrugated and combined into a mosaic through an equalization and knitting procedure that ensured continuity along the edges of adjacent areas [27] using the software listed in Table 1. The magnetic field intensity grid obtained by this procedure was used to create a magnetic anomaly grid according to the method developed by Reference [26]. In particular, the magnetic anomalies $\Delta T$ were calculated by subtraction of a 3th degree polynomial, as follows:

$$\Delta T(x, y; N) = T(x, y) - \sum_{n+m \leq N} a_n b_m x^n y^m \tag{4}$$

where $T(x,y)$ are observed total field intensities, $N = 3$ is the polynomial degree of the reference field, and the coefficients $a_n$ and $b_m$ were calculated by statistical regression on the observed data. The uncertainty, $\varepsilon$, associated with the magnetic anomaly field obtained by Equation (4) was calculated by the following expression [26,27]:

$$\varepsilon(x, y) = \varepsilon_p^2 |\nabla T| / 2^{1/3} + \varepsilon_0 \tag{5}$$

where $|\nabla T|$ is the analytic signal of the observed total field intensities, $\varepsilon_P$ is the maximum estimated positioning error, and $\varepsilon_0$ is the background uncertainty associated with the statistical regression of the observed total field intensities to a polynomial surface [26]. On the basis of the field conditions, we estimated a maximum positioning error $\varepsilon_P = 0.15$ m, while the regression uncertainty resulted to be $\varepsilon_0 = 5.29$ nT.

## 2.5. Paleomagnetic Acquisition & Processing

We sampled 13 cores from the Pozzolanelle unit using a portable drill. The cores were oriented using a magnetic compass. The sampling was integrated by soil specimens of the substratum in the survey area. The measurements were carried out in the paleomagnetic laboratory of ISMAR–CNR in Bologna. The NRM of the tuff was acquired by these rocks at the time of their deposition and cooling. Therefore, it is an expression of the Earth's magnetic field direction at that time. The induced magnetization of the tuff and soil units depends on their magnetic susceptibility and was directed as the Earth's magnetic field at survey time in the Plutonium-Inferi area. Both the tuff NRM and the magnetic susceptibility of the tuff and soil units represent fundamental quantities that will be used in the subsequent magnetic field modelling procedure. In order to obtain the characteristic remnant magnetization (ChRM) of the rock unit, a stepwise alternating field (AF) demagnetization process was applied to the collected samples. The bulk volume susceptibility, $\chi$, was measured using a Bartington MS2 susceptibility meter and the results were used to calculate the induced magnetization.

To determine the magnetic minerals that are responsible for the tuff remnant and induced magnetization components, we applied a Mössbauer spectroscopy technique. This is a powerful method that can be used for the identification of iron-bearing minerals when the magnetic properties of the material are due to a mixture of ferromagnetic minerals. In particular, this technique is very useful when the sample contains minerals with similar magnetic properties, not easily distinguishable using traditional rock magnetic methods, or when weakly magnetic minerals like hematite are mixed with minerals characterized by stronger magnetization, for example magnetite [28].

### 2.6. Electric Resistivity Acquisition & Processing

Electrical resistivity data were acquired using a Geopulse Tigre 128 resistivity meter as a support technique, with the objective of assessing the existence of structures that had already been detected on the basis of magnetic or GPR methods. The acquisition parameters were set to optimize the input current, the shape of the current signal input, the number of averaged potential readings, and the sampling time. This instrument is able to automatically cancel the spontaneous potentials that are eventually generated in the ground and that, in the case of small drops in potential during the measurements, would have been one of the major sources of noise. Before carrying out the measurements, a control test was carried out on the value of the contact resistance between the electrodes and the ground, accepting a resistance value lower than 2000 $\Omega$. We performed ten resistivity acquisitions using 32 electrode spreads. Eight 2D geoelectric profiles out of ten were created using a Wenner–$\alpha$ geometry and 1 m electrode separation, while the remaining two profiles were acquired using a dipole-dipole configuration with 2 m electrode separation. The former array is more sensitive to vertical changes in subsurface resistivity below the center of the array and less sensitive to horizontal changes in the subsurface resistivity. The advantage is that a smaller amount of acquisitions are necessary to build a pseudosection and a more precise determination of the depth to voids can be accomplished with this technique [29]. In addition, Wenner–$\alpha$ arrays display a higher S/N ratio, while initial tests showed that the dipole-dipole configuration did not provide a significantly better detail in the resistivity field maps. For each profile, the repetition time interval of the square sampling wave was set to 2.8 s, while the number of averaged readings for the estimate of the resistivity was set to 4. The resulting data set was processed using the software listed in Table 1 and a search criterion based on the minimum data heterogeneity [30]. In the initial pre–inversion step, we analyzed the data quality of each electric resistivity profile through a test inversion and displayed the resulting root mean square (rms) errors. The subsequent inversion step adopted a strategy based on a 10% increase in the thickness of the layers as the depth increased.

### 2.7. Aerial Photogrammetry

Building a magnetization model of the study area through forward modelling techniques requires a precise DEM of the area where the magnetic data were collected. We used unmanned aerial vehicle (UAV) photogrammetry to create a DEM of the Plutonium-Inferi complex (Figure 8), according to the Structure from Motion technique. The UAV was a DJI Phantom 4 equipped with an RGB visible light camera (1/2.3″ CMOS, 12.35 M effective pixels, 12.71 M total pixels, lens FOV 78.8° 26 mm f/2.2, distortion <1.5%, focus from 0.5 m to ∞), flying at 40 m altitude at ~4 m/s speed. A GPS supported by the regional network was used for georeferencing a set of ground control points and to scale the aerial photogrammetric survey. The bare Earth extraction from the DSM was accomplished using the software listed in Table 1 and a point clouds segmentation and classification algorithm. The final DEM covered an area of 30,230 $m^2$ with a pixel size of 0.0456 × 0.0456 $m^2$.

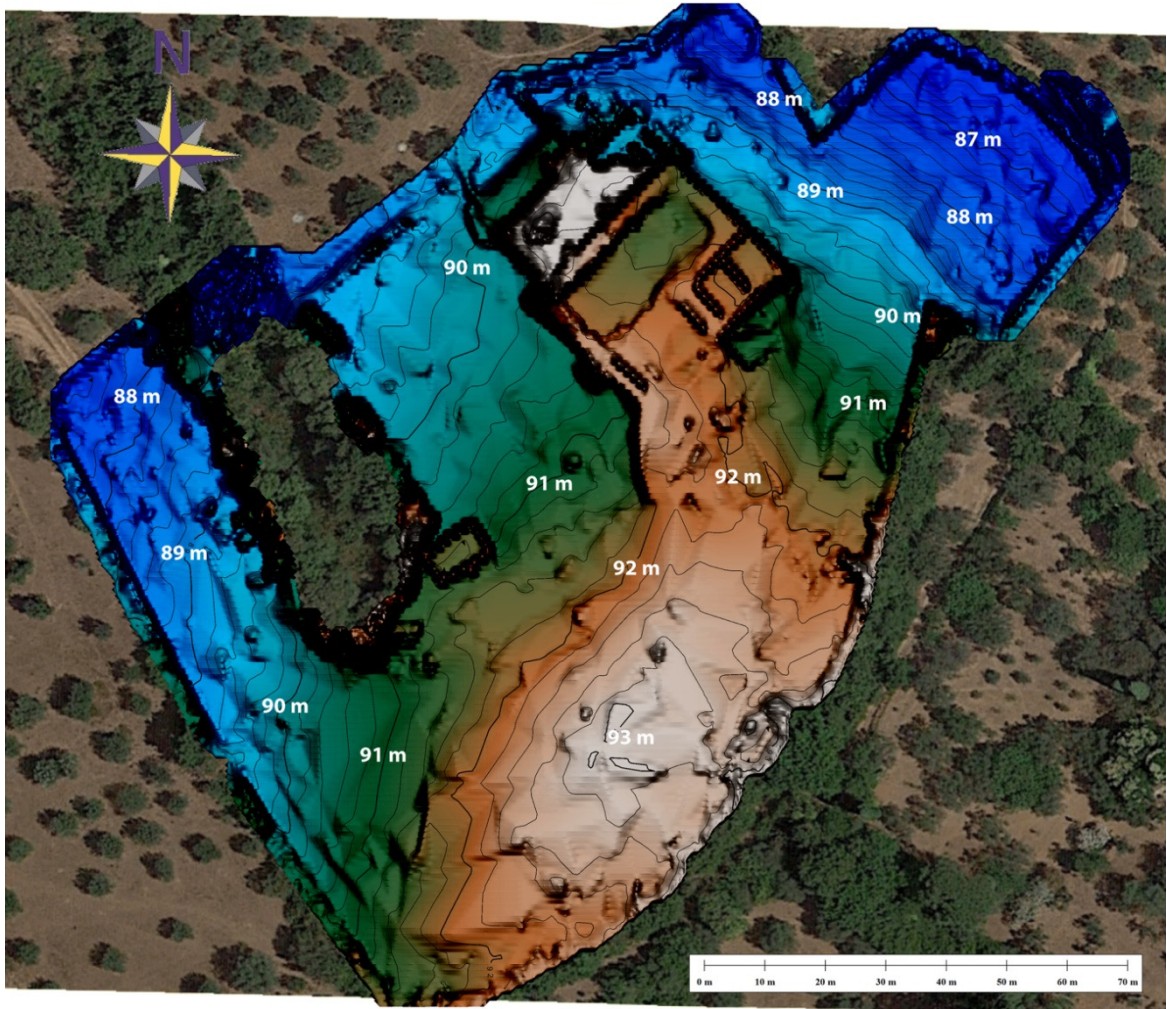

**Figure 8.** DEM of the study area (Plutonium-Inferi complex), based on aerial photogrammetry. Contour lines are spaced 0.25 m.

## 3. Results

### 3.1. GPR Amplitude Slices

We compiled four depth slices corresponding to the intervals 45–74 cm, 120–146 cm, 162–187 cm, and 238–262 cm. These are shown in Figure 9. In these maps, the presence of reflectors is indicated by brown colors while their absence is shown in green or blue. The color intensity is always proportional to the corresponding reflection amplitude. However, it is worth noting that the nature of the reflectors cannot be established by the mere visual inspection of these depth slices. It can only be determined after an analysis of individual radar profiles. The interpretation of amplitude slices is complicated by the fact that structures (and associated reflectors) that are inclined with respect to the Earth surface may be subdivided between different slices and slightly displaced with respect to each other. Excluding long reflectors associated with the contact between stratigraphic units, in general, strong localized reflections are produced by the following: (1) Cavities, (2) water-saturated soil, (3) walls and other building remains, (4) modern public utilities, and (5) modern artifacts [31]. However, significant reflections may also be locally associated with partial decoupling of the antenna (for example, as a consequence of a ground surface irregularity), scatterers (e.g., stones), and nearby metals (e.g., fences). All the radar profiles in the data sets 01 through 13 (Figure 6) were analyzed individually to give a precise characterization of the sources of each relevant feature detected on the amplitude slices.

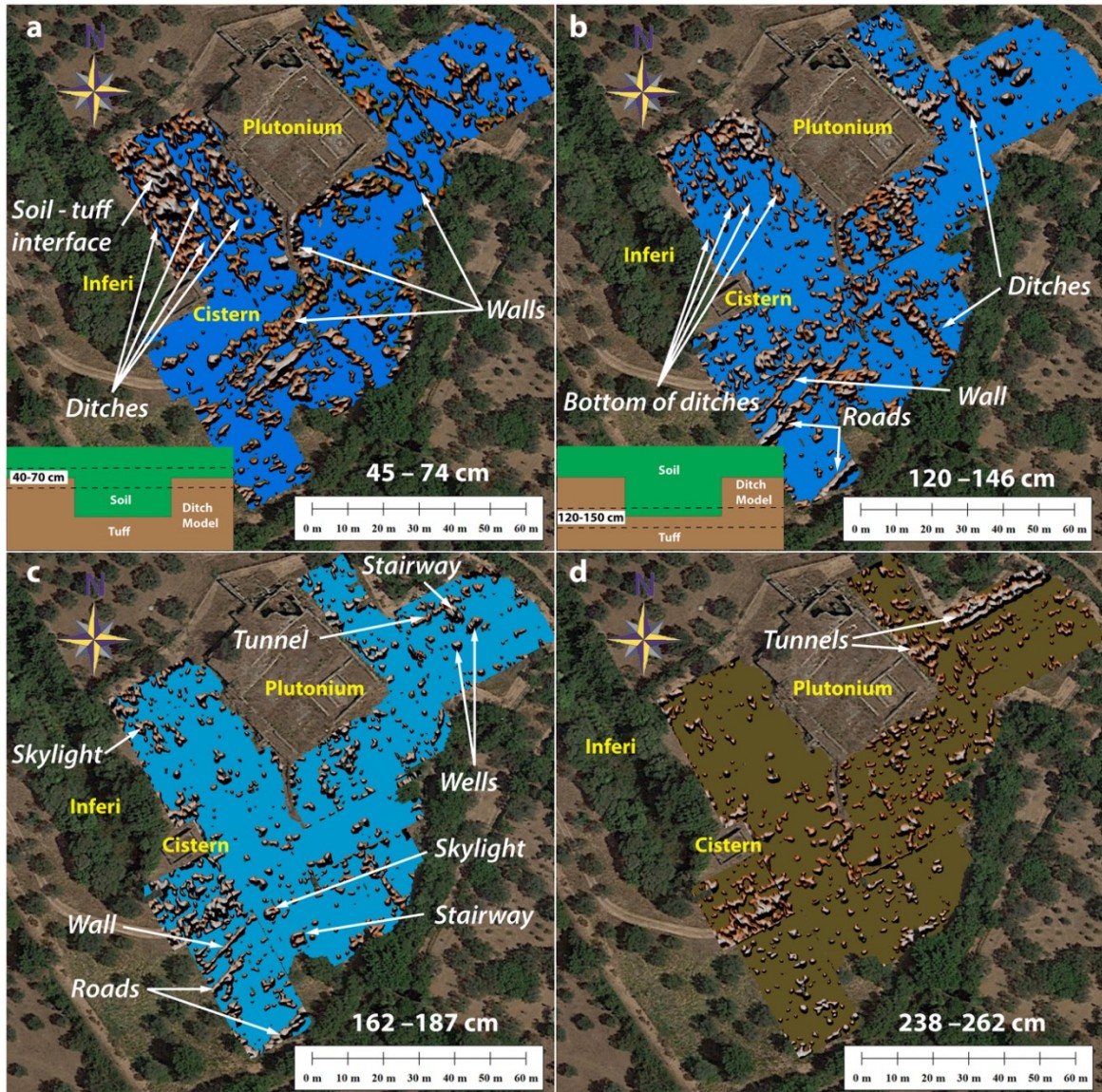

**Figure 9.** GPR amplitude maps for the depth intervals (**a**) 45–74 cm (12–20 ns), (**b**) 120–146 cm (33–41 ns), (**c**) 162–187 cm (47–55 ns), and (**d**) 238–262 cm (72–80 ns). The interpretation of strong reflection amplitudes in terms of archaeological features is discussed in the text.

An interesting feature of the 45–74 cm depth slice (Figure 9a) is represented by linear regions characterized by the lack of reflections. An analysis of reflection profiles through these regions shows that they correspond to ditches excavated at the top of the tuff layer, according to the conceptual model shown at the bottom of Figure 9a. This interpretation is in agreement with the 120–146 cm slice (Figure 9b), which shows the base of the ditches as linear reflectors. Both the 45–74 cm and 120–146 cm amplitude slices show the presence of few structures that can be interpreted as walls or roads, but a complete analysis of these shallow features goes beyond the scope of this paper. The next 162–187 cm and 238–262 cm depth slices (Figure 9c,d) show the first evidence of structures that can be interpreted as tunnels and skylights. This interpretation will be discussed in the next section.

*3.2. Magnetic Anomalies*

Figure 10a shows the magnetic anomalies observed in the Plutonium-Inferi area. These anomalies have very strong intensity and are mostly due to air- and soil-filled cavities embedded in the highly magnetized bedrock. In particular, the strong circular negative anomalies visible in the map are the

magnetic expression of the buried skylights observed on the GPR amplitude slices (Figure 9c,d), while the narrow linear negative anomalies reveal the presence of the ditches excavated in the tuff layer (Figure 9a,b). Differently from the maps in Figure 9, the unique visual evidence of tunnels in the magnetic anomalies consists into a small 20 m stripe of negative amplitudes aligned with the sequence of skylights (Figure 10a).

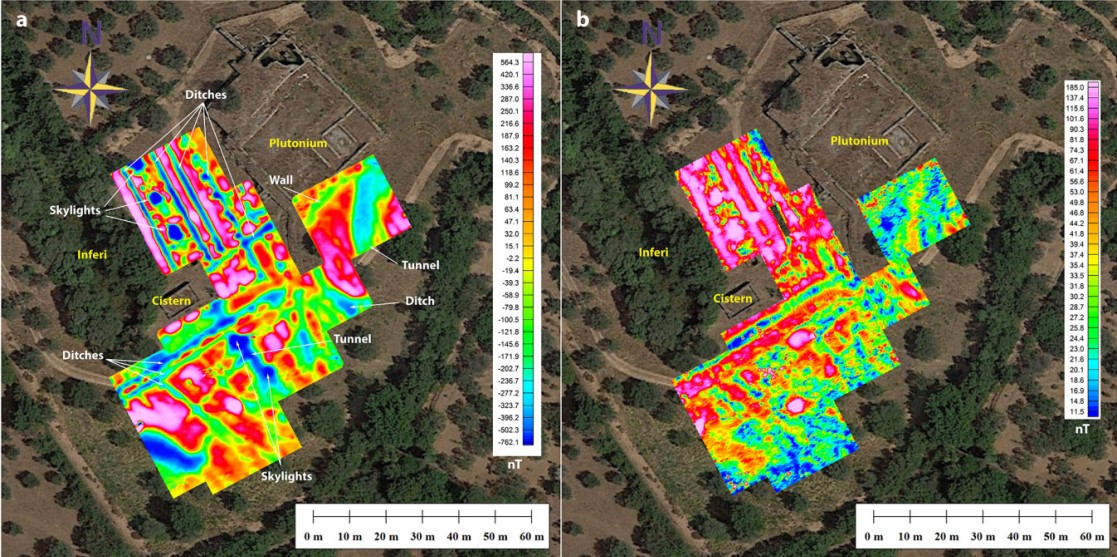

**Figure 10.** (**a**) The magnetic anomaly map of the Plutonium-Inferi complex; (**b**) Uncertainty grid.

The uncertainty associated with the observed anomalies is shown in Figure 10b. This map shows that the highest uncertainty values are found where the magnitude of the anomalies exceeds 1000 nT. An estimate of the uncertainty associated with the total field observations is necessary for the modelling of the magnetic sources (see below) to avoid the overfitting of magnetic anomalies that have been calculated on the basis of a magnetization model to anomalies that are associated with observed total field intensities at a level of accuracy exceeding the actual uncertainty.

We also generated the radially averaged power spectrum [32] of the magnetic anomaly grid, with the primary objective of checking that the ensemble with the highest slope had a depth compatible with the maximum expected depths of the archaeological features [26,27,32]. However, this kind of analysis also provides a quantitative estimate of the average depths associated with the statistical ensembles that constitute the magnetic sources, which will be used in the subsequent procedure of forward modelling. The radially averaged power spectrum of Figure 11 shows the existence of four ensembles at different depths, which are in perfect agreement with the observed archaeological features in the Plutonium-Inferi area. The deepest set of sources can be found at depths exceeding 2.6 m and is most probably due to the presence of tunnels, while ditches dug at the surface of the tuff could be represented by the ensemble at ~1 m (Figure 11). Finally, walls and other remains of archaeological features that are present at shallower depths (<0.5 m) are represented by the two ensembles at 0.42 m and 0.26 m (Figure 11).

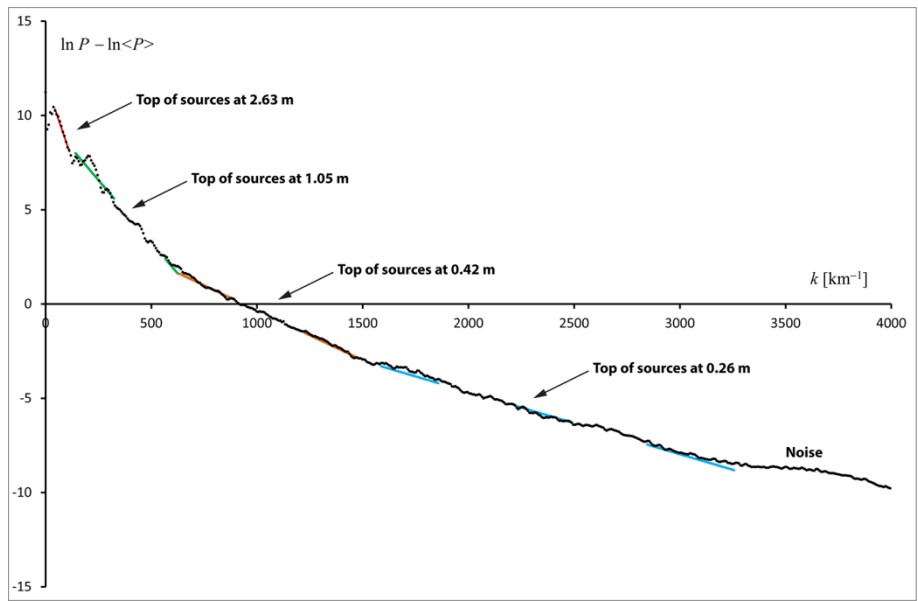

**Figure 11.** Radially averaged power spectrum of the magnetic anomalies in Figure 10a.

### 3.3. Paleomagnetic Analysis

The absorption spectrum of the Pozzolanelle ignimbrite is shown in Figure 12. The fit to these data is dominated by three strong paramagnetic doublets, but also includes a broad central singlet indicative of the presence of small grains of superparamagnetic iron oxides above their blocking temperature. Finally, the observed spectrum shows four weaker ferromagnetic sextets. The Mössbauer parameters of the doublets are characteristic of $Fe^{+3}$ or $Fe^{+2}$ in octahedral coordination, as it is often observed in the spectrum of biotites, but much more likely arise from the presence of superparamagnetic iron oxyhydroxides, such as goethite or ferrihydrite, which are common alteration products in volcanic rocks. These components account for 64% of the spectral area, while another 13% is represented by the superparamagnetic iron oxides. The remaining four ferromagnetic components suggest the presence of maghemite (9%), non-stoichiometric magnetite (7%), and, possibly, superparamagnetic hematite just below the blocking temperature (8%). Although the interpretation of the latter component is only a tenable hypothesis, it is supported by the large amount of hematite visible on the sample (see Figure 3) and by a preliminary X-ray diffraction analysis performed on the specimen.

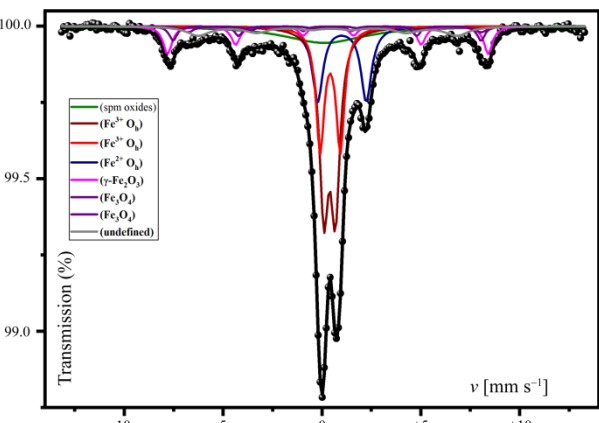

**Figure 12.** Room temperature $^{57}$Fe Mössbauer spectrum of a Pozzolanelle unit specimen. Dots are original data points and the black line is the best fitting envelope of the eight spectral components.

The paleomagnetic laboratory analysis performed on 13 sampled cores showed that the Pozzolanelle ignimbrite exhibits a single component of magnetization with only a minor viscous magnetization, easily removed after 5 mT of AF field (Figure 13a). The results listed in Table 3 show that these rocks have strong NRM and a relevant induced magnetization, in agreement with the results of Mössbauer spectroscopy. From these data, we obtained an average volume magnetic susceptibility $\chi$ = 18,126.92 × 10$^{-6}$ SI units. Therefore, assuming a reference field vector $F$ with an intensity $F$ = 46,489.2 nT and direction ($D_0$ = 3.14°, $I_0$ = 58.16°) on 9 February, 2018 at 41.937567°N, 12.779273°E, results in a mean induced magnetization $M_I$ = ($\chi/\mu_0$)$F$ = 0.67 A m$^{-1}$. The average NRM intensity results to be $M_R$ = 4.82 A m$^{-1}$ (13 specimens mean) with a mean direction of ($D$ = 4.1°, $I$ = 72.8°) (10 samples average) and the statistical parameters $k$ and $\alpha_{95}$ as $k$ =141.3 and $\alpha_{95}$ = 4.1° (Figure 13b), respectively. This result is statistically compatible with a previous study [12]. Taking the rapid deposition and cooling of the pyroclastic flow into account, this paleomagnetic direction can be considered as representative of the geomagnetic field at the time of deposition of the Pozzolanelle unit.

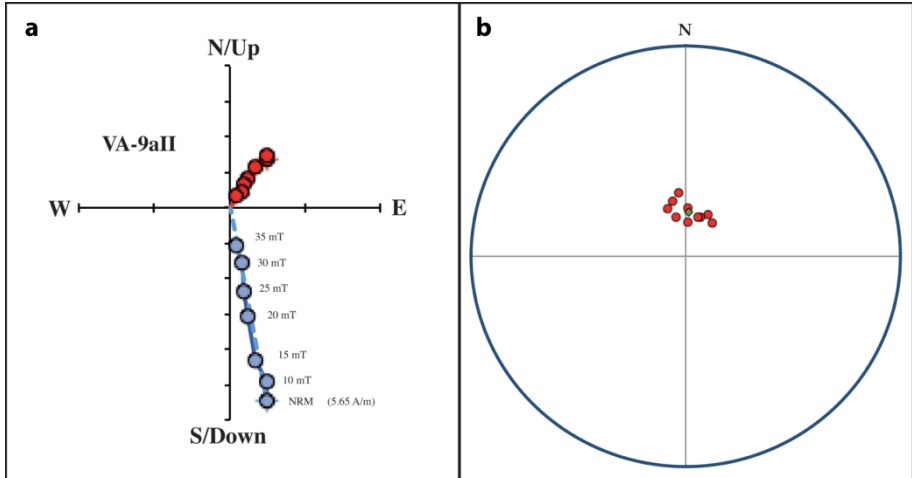

**Figure 13.** (**a**) AF demagnetization plot and (**b**) paleomagnetic directions (red dots) with resulting mean (green dot).

**Table 3.** Results of palaeomagnetic analysis.

| Sample Id. | $D_{PCA}$ [°deg] | $I_{PCA}$ [°deg] | MAD [°deg] | NRM [A m$^{-1}$] | $\chi$ [×10$^{-6}$] | Q |
|---|---|---|---|---|---|---|
| VA–4 | 346.72 | 74.31 | 0.88 | 4.58 | 18,820 | 6.98 |
| VA–5 | 2.63 | 71.10 | 1.06 | 4.71 | 19,170 | 7.05 |
| VA–6aI | 27.87 | 71.73 | 0.77 | 4.81 | 18,720 | 7.38 |
| VA–6aII | 38.54 | 73.77 | 0.98 | 5.05 | 18,850 | 7.68 |
| VA–7aI | 347.26 | 68.20 | 0.68 | 4.95 | 17,140 | 8.29 |
| VA–7aII | 353.76 | 65.32 | 0.48 | 5.30 | 19,410 | 7.83 |
| VA–7aIII | 19.76 | 73.96 | 0.62 | 4.84 | 18,360 | 7.56 |
| VA–8 | 339.25 | 70.30 | 1.29 | 5.00 | 18,250 | 7.86 |
| VA–9aII | 16.47 | 74.10 | 0.63 | 5.65 | 18,240 | 8.88 |
| VA–9aI | 2.97 | 76.88 | 1.10 | 4.24 | 14,980 | 8.12 |
| VA–1 | | | | 4.07 | 14,800 | 7.90 |
| VA–2 | | | | 5.04 | 22,290 | 6.49 |
| VA–3 | 29.59 | 70.48 | | 4.42 | 16,620 | 7.63 |

$D_{PCA}$, $I_{PCA}$ = NRM declination and inclination from principal components analysis. MAD = Maximum angular deviation from mean. $Q$ = Koenigsberger ratio.

### 3.4. Electric Resistivity Survey

A series of ten ERT profiles were carried out with the objective of assessing the existence of features already detected by magnetic or GPR methods. Their location in key sectors for the reconstruction

of the tunnel network around the Plutonium-Inferi area is shown in Figure 14. The data processing consisted of the inversion of the observed apparent resistivities accompanied by a severe reduction of side effects. The results illustrated in Figures 15 and 16 generally show the presence of strong resistivity contrasts, sometimes associated with sharp resistivity gradients. The moderately high absolute resistivities suggest that probable cavities present in the subsoil could be partially filled with sediment.

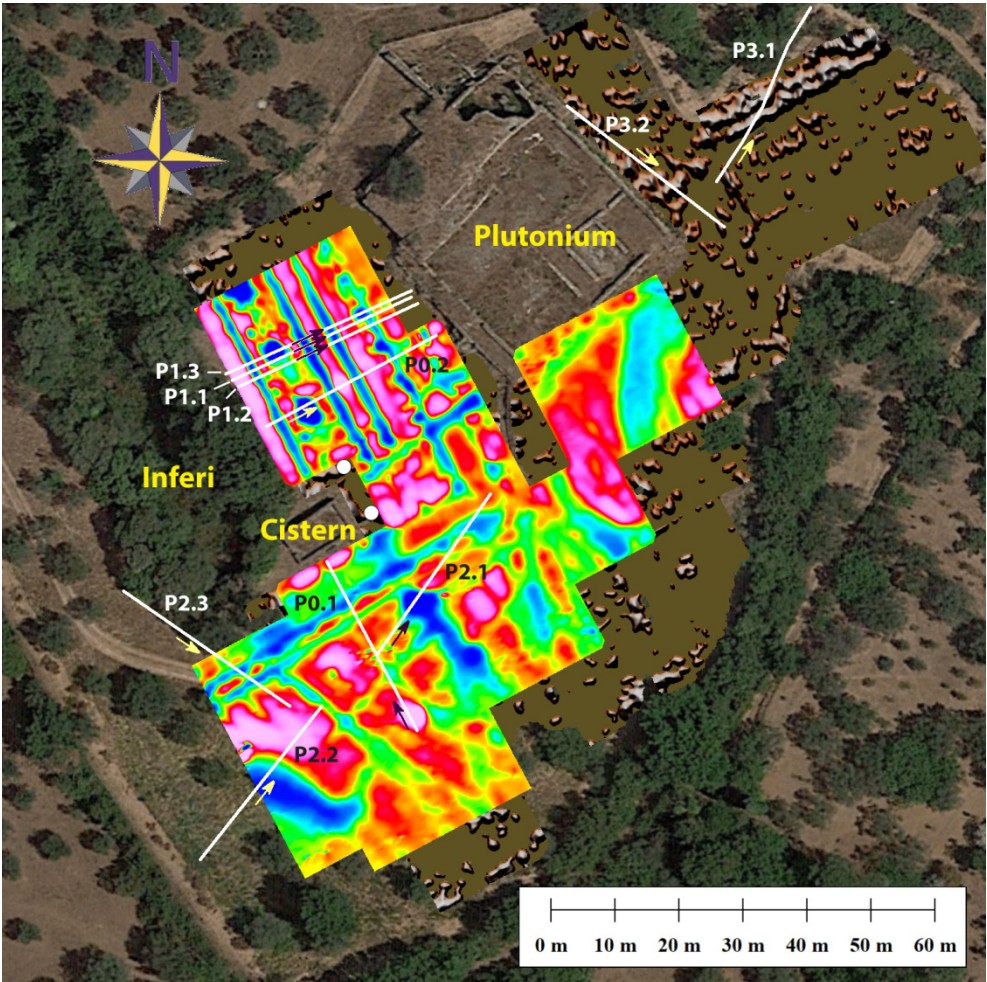

**Figure 14.** Location of ERT profiles deployed in the Plutonium-Inferi area. The background images show magnetic anomalies and the GPR amplitude slice at 238–262 cm (72–80 ns). White circles are open skylights of the main tunnel across Area 1.

All the profiles show a predominant electric layer with resistivity values in the order of tens of $\Omega$ m (on average 50–70 $\Omega$ m), which can be attributed to the tuff underlying a heterogeneous anthropogenic soil of variable thickness not exceeding 0.5 m. Four ERT profiles acquired in Area 1 (P0.2, P1.1, P1.2, and P1.3) (Figure 14) show the presence of a resistivity body (>150 $\Omega$ m) characterized by strong contrast with respect to the surrounding regions. In particular, profile P1.1 shows a noticeable resistivity anomaly, interpreted as a skylight, which is aligned with two nearby open skylights of the Strada Carrabile, one of which is shown in Figure 4. This observation allows us to infer that moderately high values of resistivity can be correctly attributed to the presence of tunnels and skylights partially filled by soil. Therefore, all the high resistivity regions in the ERT profiles of Figure 15 are interpreted as filled cavities. Similarly, the resistive body observed in profile P2.1 (Figure 16), which is also aligned with the main tunnel, is interpreted as a skylight cavity above this tunnel. Conversely, profile P2.2 (Figure 16) shows a different situation, characterized by the presence of a low resistivity sub-horizontal

layer ($\rho$ < 20 $\Omega$ m) and strong resistivity gradients, whose base deepens from shallow depths down to about 3 m and can be associated with a wet detritus set on an erosion morphology of the tuff bank, possibly associated with a bench terrace.

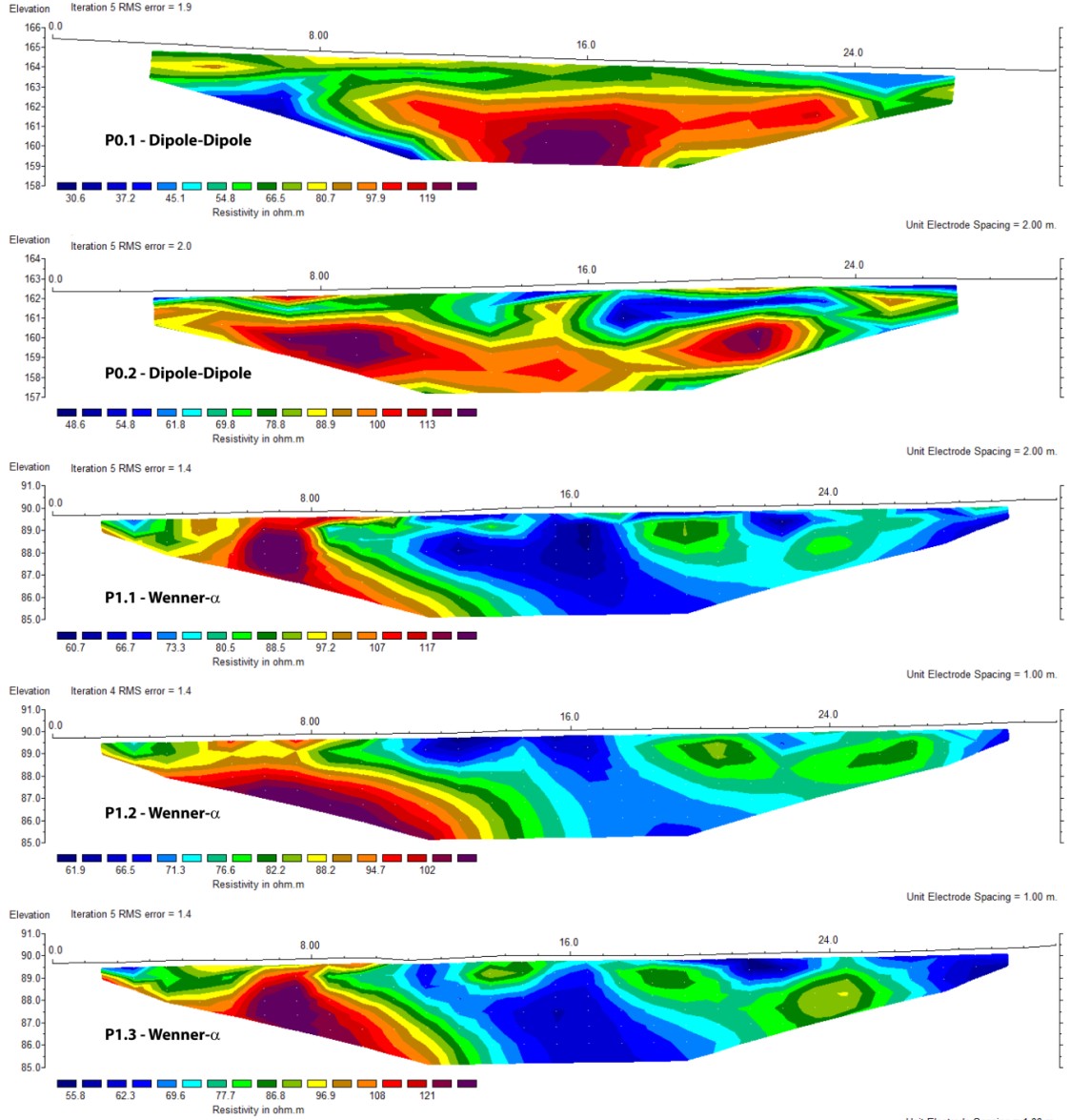

**Figure 15.** ERT profiles in Areas 1 and 6. Their location is shown in Figure 14.

All the remaining profiles of Figure 16 show prominent resistivity anomalies (>120 $\Omega$ m) that can be interpreted as tunnels. However, in the case of profile P2.3 we have no independent data source that can confirm the existence of a tunnel, except that a tunnel in this location and with a compatible orientation is included in one of the original Piranesi's maps as a diverticulum of the Grande Trapezio [33]. Therefore, further studies will be necessary to confirm the existence of a tunnel in this area. Similarly, the resistivity data of profile P0.1 (Figure 15) show a well-localized anomaly (>130 $\Omega$ m) at a depth of over 3 m and placed in the central part of the profile, which can be interpreted as a tunnel. This anomaly could be related to that of profile P2.3 and be representative of the same structure. However, in this case we have no other data source that can confirm this hypothesis, because the penetration depth attained by our radar survey did not exceed ~2.5 m.

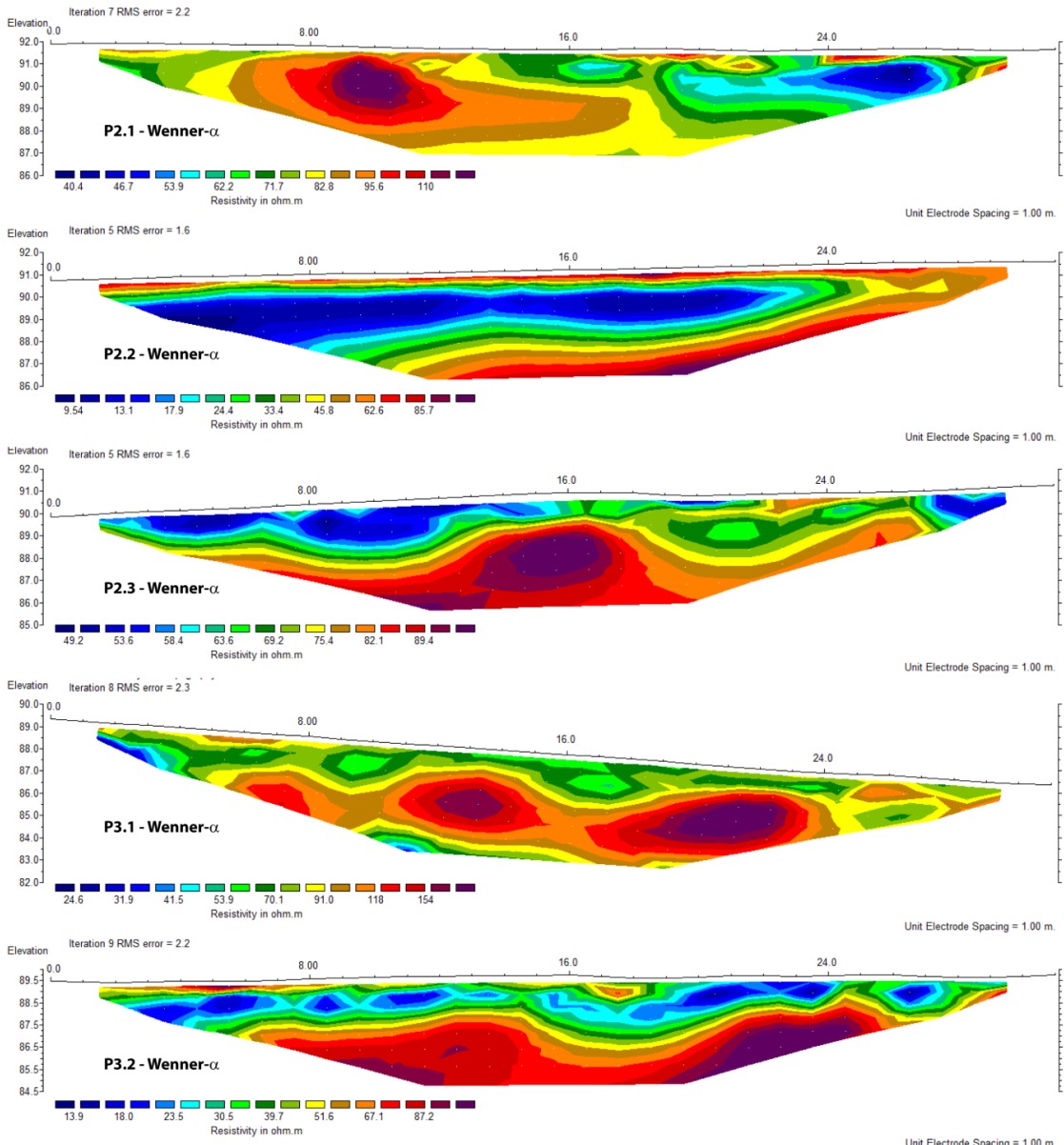

**Figure 16.** ERT profiles in Areas 2, 6, M21, 8, and 10. Their location is shown in Figure 14.

Finally, two resistivity profiles (P3.1 and P3.2, Figure 16) were deployed in areas 8 and 10 (Figure 14) to support the detection of tunnels by GPR methods in this part of the Plutonium-Inferi Complex. In the case of the P3.1 profile, the results show two distinct resistivity anomalies (>170 Ω m), centered around 13 m and 21 m, well distinct from the less resistive context that can be associated with the tuff basement, which are observed starting from a depth of around 2 m. They support the hypothesis of a couple of parallel tunnels running in ENE–WSW direction towards the Plutonium. A third moderately high resistivity anomaly can be observed close to the left edge of P3.1 (~7 m), which is compatible with the existence of a narrow conduit. The resistivity data in profile P3.2 (Figure 16) show shallow moderately resistive material (>50 Ω m) overlying a more conductive surficial anthropogenic layer (<25 Ω m) and a pronounced horizontal resistivity anomaly with maximum values at depths greater than 2 m. The high resistivity region can apparently be separated in two parts, centered respectively at 12 m and 23 m, which could be interpreted as the continuation of the tunnels detected on profile P3.1.

## 4. Discussion

We performed a quantitative modelling of the observed magnetic anomalies [26,27], with the primary objective of assigning precise locations to tunnels and ditches in the Plutonium-Inferi area of Hadrian's Villa. As mentioned above, Hadrian's Villa lies on a substratum composed by an ignimbrite tuff massive deposit with very high magnetic susceptibility $\chi$ and strong NRM. In this area, the tuff is also covered by a layer, ~0.3 to 1 m thick, of very magnetic soil with a susceptibility of $\chi_0 = 9500 \times 10^{-6}$ SI units. The magnetization model shown in Figure 17 does not include most of the archaeological features buried in this topsoil layer. Therefore, the fit between the calculated and observed anomalies is rather coarse, although it explains the high-amplitude anomalies observed in this area. These two layers were modelled as slabs underneath the whole area at depths 0.5–5 m and 0–0.5 m respectively. To model empty tunnels and skylights in the tuff, we used rectangular prisms and cylinders, respectively, embedded in the tuff unit and with opposite magnetization parameters, as follows: $M_R = -4.82$ A m$^{-1}$, $D = 4.1°$, $I = 72.8°$, $\chi = -18127 \times 10^{-6}$ SI units. Similarly, to create a model of soil-filled ditches carved in the tuff and soil-filled tunnels, we defined vertical prisms that overlapped the tuff layer with opposite NRM and a susceptibility $\chi = \chi_0 - 18127 \times 10^{-6}$ SI units $= -8627 \times 10^{-6}$ SI units.

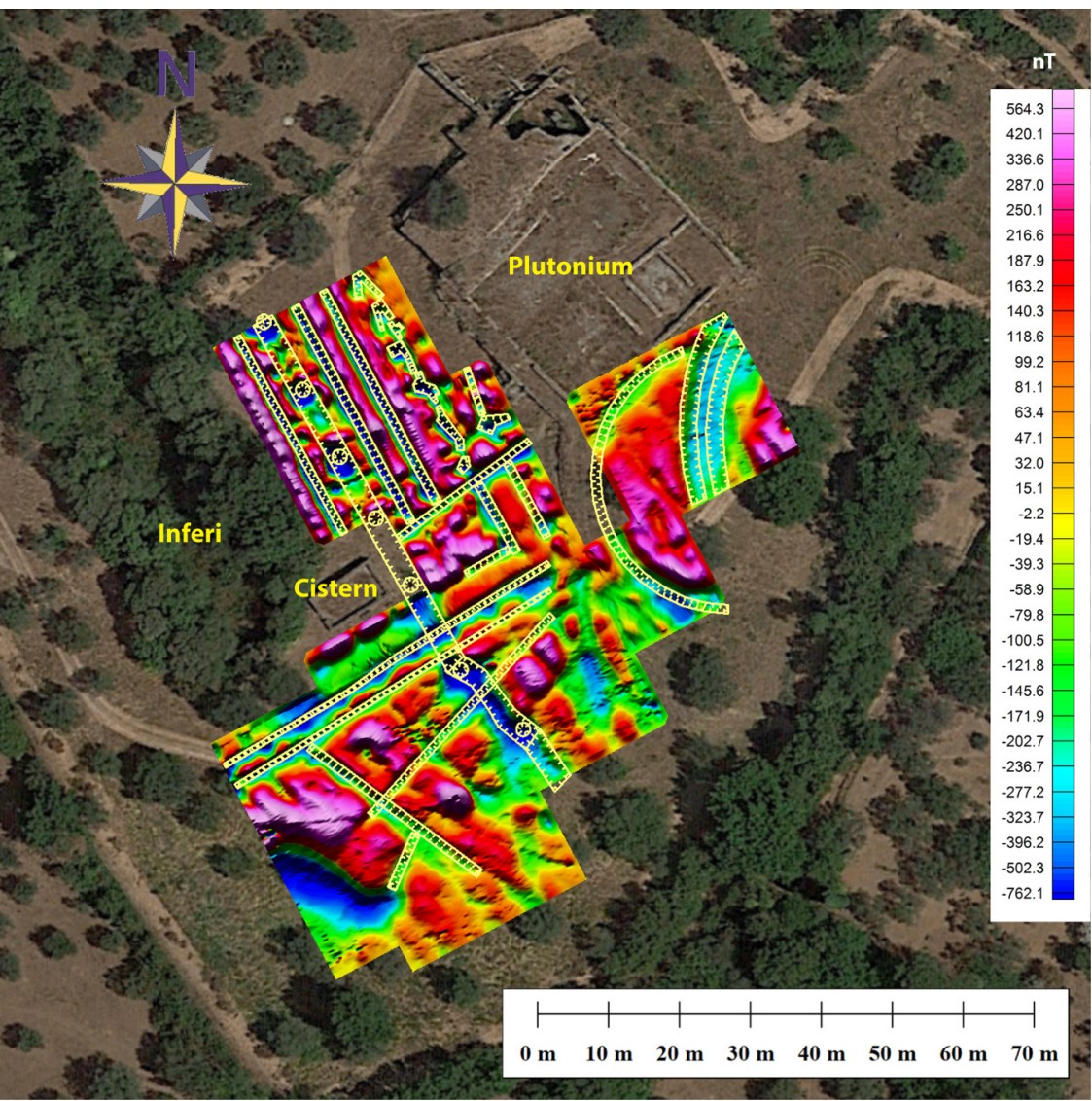

**Figure 17.** Magnetization model (yellow lines with barbs) of the tunnels, skylights, and ditches detected in the Plutonium-Inferi area on a shaded magnetic anomaly map.

The integration of the different data sources presented in the previous section allowed us to draw a realistic layout of the tunnel network beneath the Plutonium-Inferi area. We used a computer-assisted procedure of forward modelling to generate a magnetization model that explained the observed anomalies in the western portion of the complex. An example of the accuracy of the magnetization model in Figure 17 in explaining the observed anomalies is illustrated in Figure 18. The very good fit between observed and theoretical anomalies in the two profiles suggests that the voids and soil-filled structures in the tuff are responsible for most of the magnetic signals in this area, while the walls and other artifacts embedded in the topsoil account only for a minor contribution to the observed data.

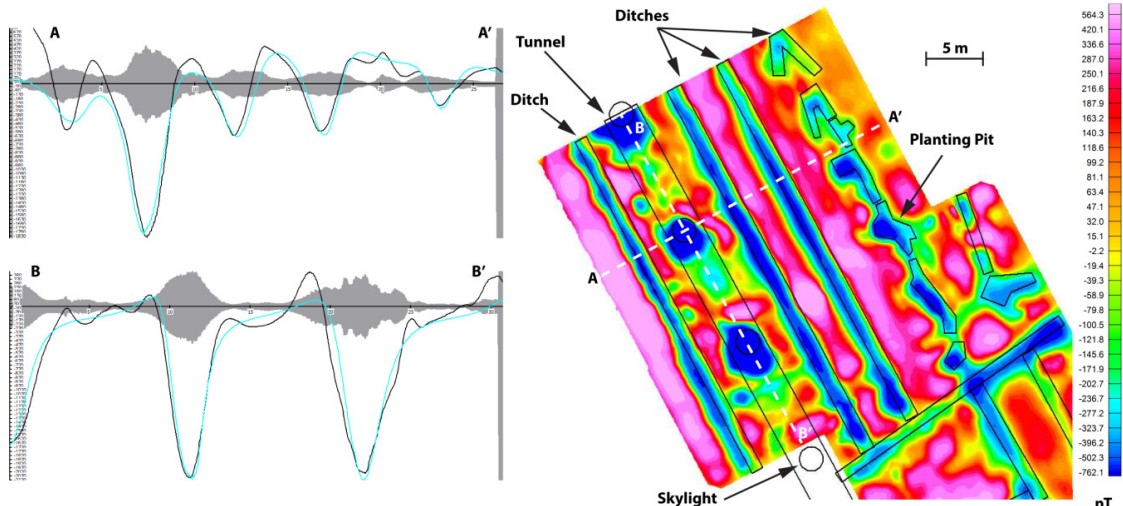

**Figure 18.** Magnetic profiles across the ditches, skylights, and the main tunnel of areas M15, M16, and M18 of Figure 7 (shapes bounded by black lines). The two profiles show observed and calculated anomalies as black and blue lines, respectively, while the grey area shows the local uncertainty.

The magnetization distribution shown in Figure 17 was constrained and complemented by GPR and electric resistivity data to overcome the intrinsic ambiguity of potential field data modelling. A meaningful example of the data integration procedure is illustrated in Figure 19, which shows a correlation between the three data sources. The GPR profile of Figure 19 shows a large hyperbole at 7 m offset, generated by a metallic lid covering the skylight [1]. The presence of a large tunnel flanking the Inferi (Figure 2) is confirmed by the clear resistivity anomaly recorded by the ERT P1.1 profile (also evident in the other profiles parallel to it, Figure 15). Although the magnitude of the high resistivity values in this ERT profile is not elevated, the direct field evidence of the presence of a large tunnel at ~2.5 m depth in this area, also indicated by two open skylights, implies that the moderately high resistivity contrast is associated with the presence of cavities partially filled by sediment. Profile P1.1 also shows shallow low resistivity regions (in blue), which are well correlated to interrupted radar reflections of the soil–tuff interface detected on the GPR profile, which are interpreted as ditches carved at the top of the tuff. Finally, the GPR profile of Figure 19 also shows the bottom of a 180 cm high void space (20–32 ns) above the soil that partially fills the tunnel.

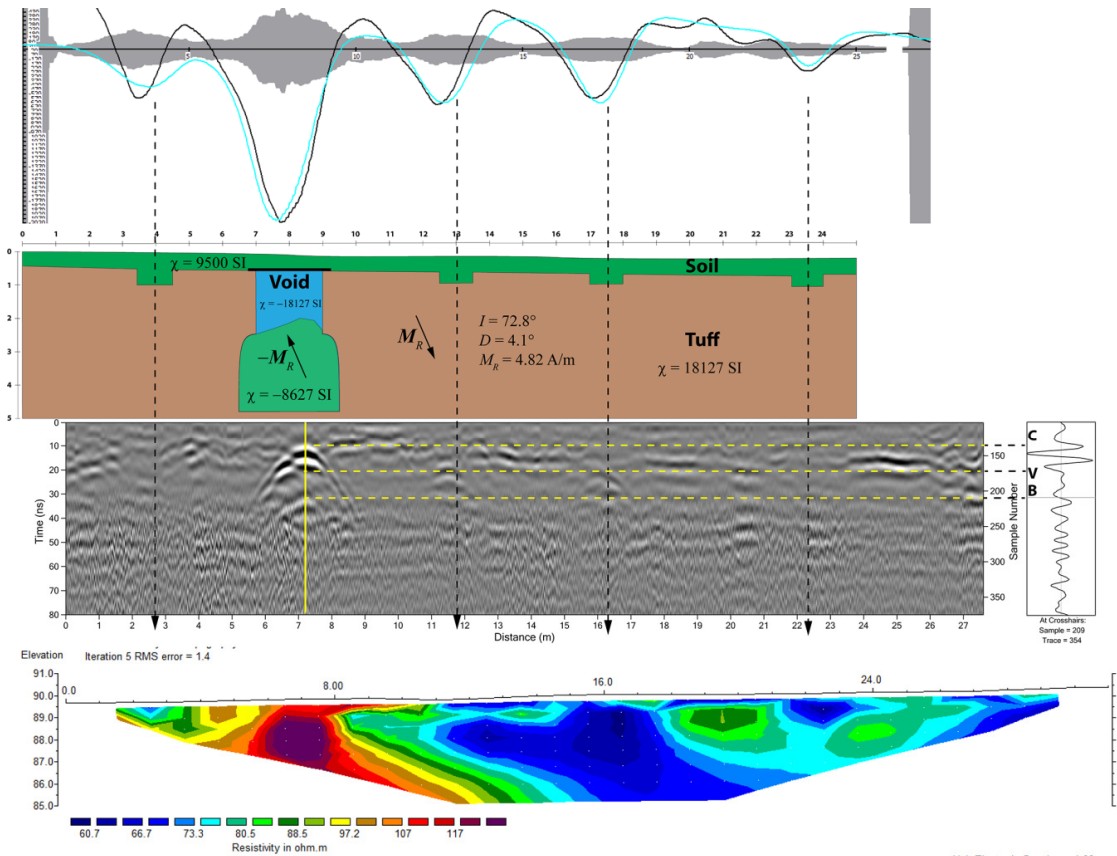

**Figure 19.** Correlation chart between magnetic, radar, and resistivity data showing ditches, skylights, and the main tunnel in areas M15, M16, and M18 of Figure 7. From top to bottom: Magnetic profile along the trace A–A' of Figure 18, conceptual magnetization model, radar profile, and projected electric resistivity profile. C = cover wavelet; V = void top wavelet; B = void bottom wavelet. The yellow line indicates the location of the A-scan to the right of the GPR profile. The black line sealing the void in the conceptual model represents a metallic cover of the skylight.

The magnetization model in Figures 18 and 19 does not necessarily provide a complete representation of the underground structures. For example, some ERT and GPR profiles parallel to the one shown in Figure 19 suggest the presence of a narrow tunnel below the irregular ditch, running parallel to the Plutonium (e.g., profile P0.2, Figure 15). Further surveys are necessary to confirm the existence of such a deep structure. Instead, the pattern of ditches illustrated in Figure 19 provides a very good representation of the system of irrigation ducts and planting pits described by Reference [34] and indicates the presence of a garden surrounding the Plutonium.

Most of the radar profiles acquired in areas 1–13 (Figure 6) are available as supplementary material (Figure S1 and Tables S1–S13). To accomplish a reliable interpretation of these profiles, we also considered the polarity of the relevant reflected wavelets. In all cases, the soil–tuff interface was characterized by an inverted polarity of the reflection (with respect to the emitted pulse, i.e., when the first dominant peak has positive amplitude), indicating that the dielectric permittivity of the tuff was higher than that of the topsoil. All the human artifacts embedded in the topsoil are interpreted as Roman walls or pathways, apart from few modern pipes that were deployed in the 1970s when the site was occupied by a camping area. The latter can be easily recognized as hyperbolae with inverted polarity, for example in Area 1 (profiles 69–104, Table S1), Area 2 (profiles 1–41, Table S2), Area 10 (profiles 1–55, Table S10), and Area 12 (profiles 23–25, Table S12). All the skylights detected on GPR profiles (as well as the three open skylights of the investigated area) have a diameter of 2 m and start at the soil–tuff interface. One of them, in Area 1 (profiles 37–39, Table S1), is very shallow and has

been covered by a metallic lid in recent times [1], as revealed by the strong reflection hyperbola with inverted polarity in Figure 19. Conversely, a ~1.4 m deep skylight in Area 2 (profile 58, Table S2), most probably has a void space just below the soil layer, revealed by a normal polarity hyperbola (Figure 20). This interpretation is in agreement with the location of the high resistivity region observed in the ERT profile P2.1 (Figure 16). In modern times, all the remaining buried skylights and wells detected in the Plutonium-Inferi area have been filled by debris and stones (profiles 35–40, 45–49, and 52, Table S10; profiles 1–3, Table S11; profiles 19–21 and 72–80, Table S12) and the soil above them appears reworked (see e.g., profile 36, Table S10).

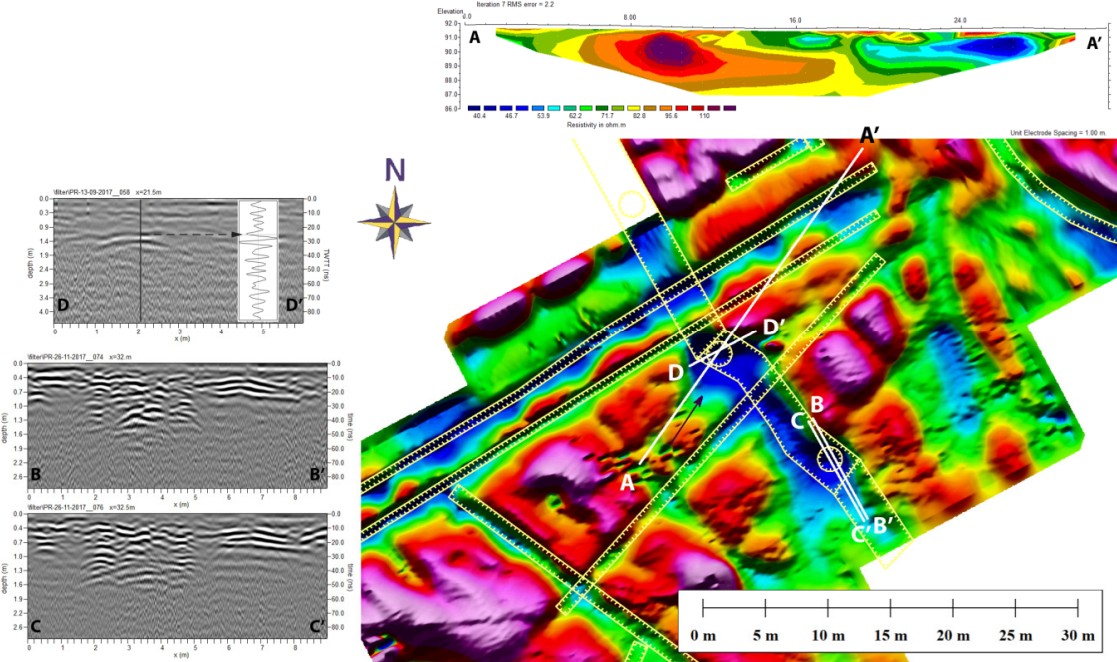

**Figure 20.** ERT and GPR evidence of skylights in areas 2 and 12 of Figure 6.

The tunnel crossing areas M15, M16, and M18 (Figure 17) correspond to the main Strada Carrabile [18] (see Figure 2). This important 5 m large underground pathway continues in SSE direction towards areas M20–M22, where it probably reaches a shallower depth, generating a strong linear negative magnetic anomaly, clearly visible along the skylights' alignment (Figure 17). Beyond the modern fence that surrounds the archaeological park, the Strada Carrabile is linked southwards to a large four-sided system of tunnels known as the "Grande Trapezio" (Figure 2) [1]. Figure 20 shows three radar profiles and the ERT profile P2.1 (Figure 16), which indirectly support the magnetization pattern associated with such tunnel. In this instance, the ERT and GPR evidence of a tunnel is only indirect, as it is given by the alignment of skylights, not by the detection of voids below the tunnel ceiling. With the exception of those identified in Area 10 (see below), all the tunnels run at a depth exceeding 2.5 m, beyond the penetration range of this survey.

The radar profiles acquired in areas 1–13 (Tables S1–S13) show evidence of several ditches dug in the tuff, which have been included in the magnetization model of Figure 17. They are revealed by an interruption of the soil–tuff interface and the downward displaced reflector (with inverted polarity) of their base. Most of these ditches are ~1 m large and reach a depth of ~70 cm in the tuff layer (Table S1), but some larger structures are also present. It is likely that part of the narrow ditches hosted lead water pipes. For example, profile 68 of Area 1 (Table S1) shows a prominent hyperbola with inverted polarity at the base of a ditch at offset $x = 16$ m, which is compatible with the recent discovery of a fistula aquaria [16]. The set of open structures excavated in the tuff also includes a 3.5 m large and 70 cm deep pool across areas 11 and 12 (Figure 6), which is cut diagonally by the radar profiles. It is especially evident on profiles 6–8 of Area 12 (see Table S12). Profile 8 also shows a structure that

can be interpreted as an intermediate step for the access to the pool. It is likely that the base of this structure was coated by reflective material, because it appears interrupted at some locations. Another important narrow ditch structure has circular shape, ~38 m diameter, and is clearly visible on the magnetic anomaly map of Figure 17. It is not easily detected on GPR profiles, because it is flanked by a circular wall that was recently excavated by the mission of Oxford and Pavia universities [16] (Figure S2). However, the presence of this ditch can be observed on many profiles of Area 3 (Table S3) and on profiles 41–43 of Area 4 (Table S4). Finally, no ditch can be observed in Area 13, where the soil–tuff interface deepens and falls below the penetration range of this survey.

The analysis of GPR and ERT profiles also revealed the existence of features not visible on magnetic maps, either because they were located outside the areas of acquisition of total field data or because of their low magnetization contrast with respect to the surrounding material. A first important finding along the Strada Carrabile was the identification of an entrance to this tunnel on a series of successive radar profiles (Figure 21). The profiles in Figure 21 (Table S12, profiles 64–68) show clear evidence of two retaining walls flanking a downgoing stairway or ramp, 1.7 m large, orthogonal to the Strada Carrabile, similar to the one described by [14] for the Accademia sector.

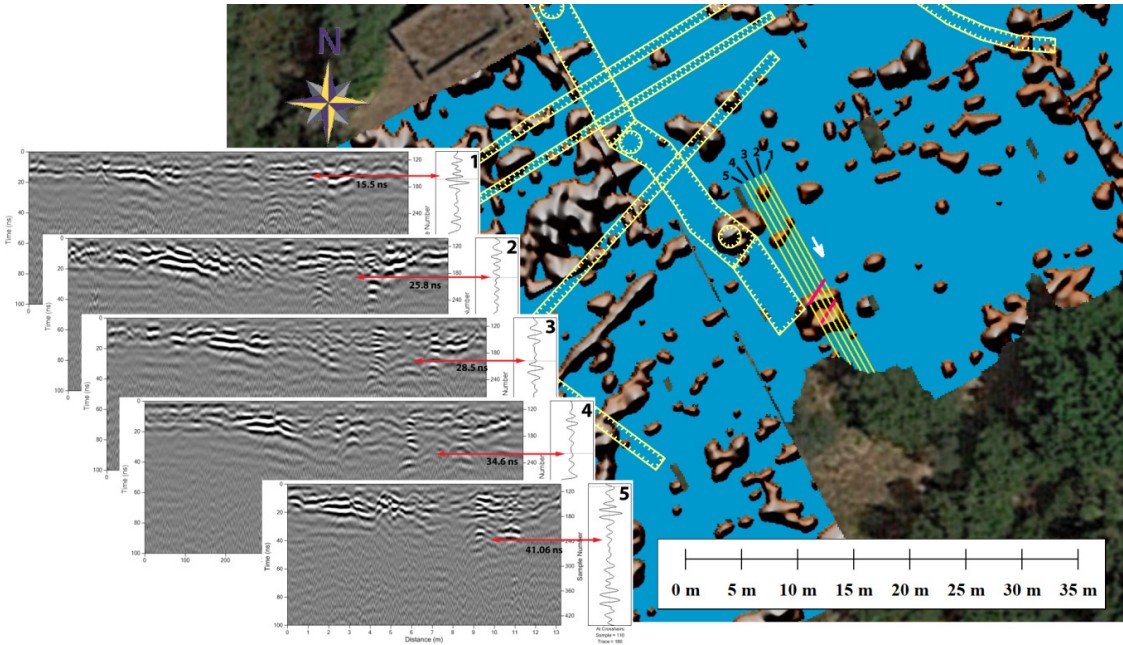

**Figure 21.** GPR evidence of a stairway entrance to the main tunnel. The existence of two retaining walls is evident on five radar profiles, which also show steps or a ramp at an increasing depth. The stairway location is indicated by the parallel red lines on the background 162–187 cm GPR amplitude slice (Figure 9c).

A second enigmatic feature was revealed by two ERT profiles (Figure 22) at a depth exceeding the range of our GPR survey. The relatively high resistivity areas in profiles P0.1 and P2.3 are compatible with cavities associated with a 3 m large tunnel transversal to the Strada Carrabile, having the ceiling at ~3 m. A tunnel with similar orientation is included on old maps of the 18th and 19th century, in one case as a diverticulum of the Grande Trapezio [33]. ERT profile P2.2 does not show any evidence of cavities, thereby it seems unlikely that a tunnel exists in the area crossed by this profile. We conclude that if a tunnel exists close to the Inferi area, it should have the orientation shown in Figure 22. A third important finding comes from the analysis of several GPR profiles from Areas 12 and 13 (profiles 60–85, Table S12; profiles 1–47, Table S13). This is a pair of parallel linear features that run in NE–SW direction towards the Plutonium, well evident on the 120–146 cm amplitude slice (Figure 23). The two features have different thickness (30 and 50 cm) and width (3 and 3.4 m, respectively) and show normal

and inverted polarities along their top and bottom reflectors. Consequently, these features are low dielectric permittivity regions that could be tentatively interpreted as basolato of Roman roads.

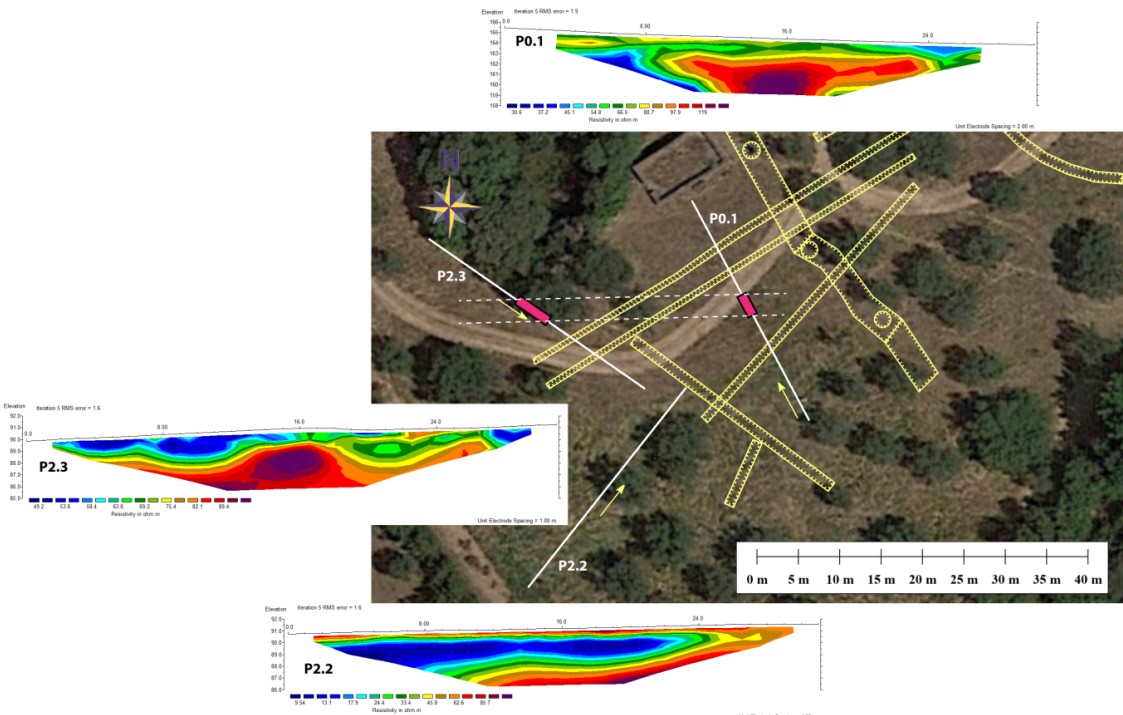

**Figure 22.** ERT evidence of a transversal tunnel. The dashed lines indicate its probable location.

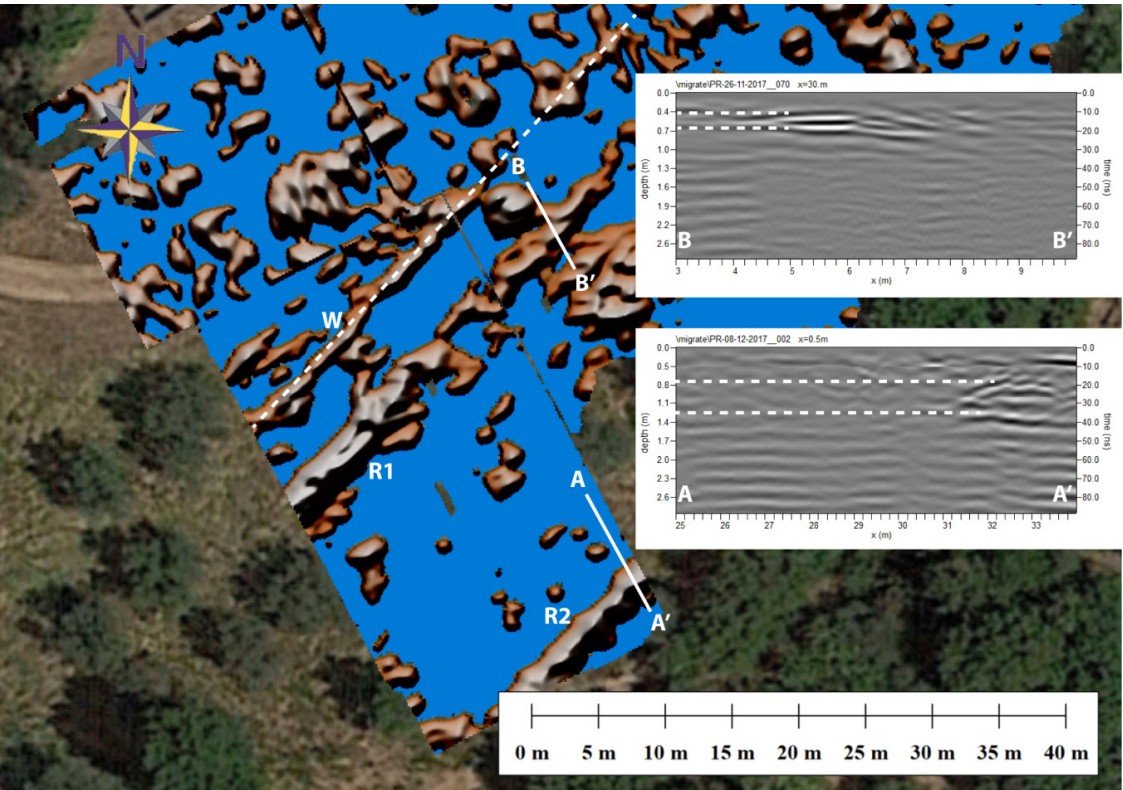

**Figure 23.** GPR evidence of two roads, R1 and R2. The two sections are migrated profiles. Feature *W* is a different archaeological structure, discussed in the text. Black and white colors show negative and positive amplitudes, respectively.

In addition to the two features interpreted as roads, Figure 23 shows another interesting archaeological structure, *W*, which is represented by a narrow linear stripe of high-amplitude reflectors, running parallel to the two roads. This is the southeastern prolongation of a wall that extends radially from the large circular ditch close to the Plutonium. The presence of this wall was also confirmed by a recent excavation [16] (Figure S2). It is easily detected on GPR profiles 81–85 of Area 12 (Table S12) and on all profiles of Area 13, up to 1.1 m depth (Table S13).

The northeastern portion of the Plutonium-Inferi complex (Areas 8, 9, and 10) has not been investigated by magnetic methods due to the presence of fences. However, both GPR and electric resistivity data revealed interesting features in this sector. The ERT and GPR profiles of Figure 24 show evidence of a couple of tunnels in the direction of the northeastern slope, which could represent a way out from the villa towards the ancient Via Tiburtina. This is the unique place around the Plutonium where we obtained a direct GPR evidence of tunnels, thanks to the shallow depth of their ceilings. In general, a tunnel ceiling is revealed on GPR profiles by a wavelet pair with normal and reversed polarities, respectively, when the cavity is air-filled. At Hadrian's Villa, these two wavelets would be generated by the tuff-air and air-soil (or air-tuff) interfaces. In the case of a soil-filled tunnel, we would record only a small amplitude normal polarity wavelet produced at the tunnel ceiling by the tuff-soil dielectric contrast. In any event, a GPR profile transversal to the tunnel strike would show only the low-curvature apical part of the ceiling, because of the very small separation between transmitter and receiver antennae. This curved shape cannot be confused with a reflection hyperbola, because only very low dielectric (say, $1 \leq \varepsilon \leq 3$) hyperbolae could be fitted against this line.

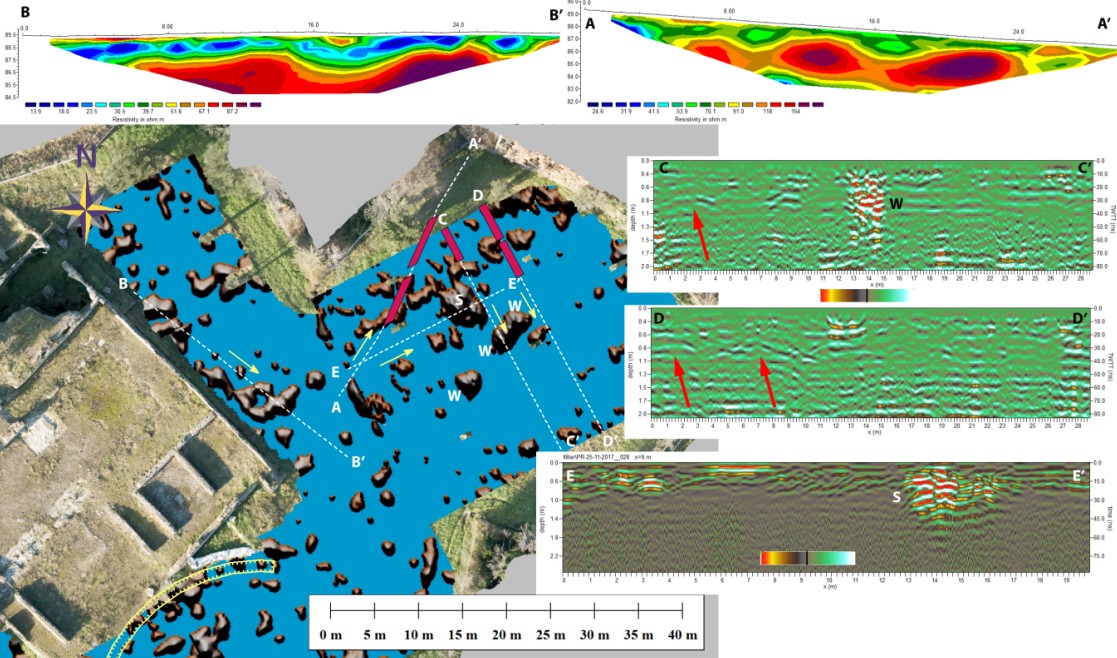

**Figure 24.** ERT and GPR evidence of a couple transversal tunnels linking the Plutonium to the northwestern slope. Features labelled W are interpreted as wells. Red arrows indicate the location of tunnels on migrated profiles C–C' and D–D'. Profile E–E' is unmigrated profile 28 (Table S10), showing a lateral open passage to the tunnels (a stairway?).

The radar profiles of Area 10 show the presence of three wells (Figure 24) filled by a jumbled assemblage of stones, easily detected on profiles 35–40 and 42–49 (Table S10). The two tunnels shown in Figure 24 had never been documented by previous studies and cannot be detected on the profiles of Table S10 due to their strike. We deployed six transversal GPR profiles (Table S10T) and two ERT profiles (Figure 16) to check the existence of tunnels in this area, because the amplitude slice maps indicated the possible presence of these features (Figure 9c,d). GPR profiles C–C' and D–D' (Figure 24)

show evidence of shallow cavities (~80 cm depth) and a geometry compatible with tunnel or conduit ceilings. This interpretation is supported by ERT profiles P3.1 and P3.2 (Figure 16), which show moderately high resistivity anomalies at the same depth (Figure 24). Another interesting feature detected on GPR profiles 16, 17, 19, 22, 25, and 27–32 of Area 10 (Table S10) is a transversal cavity, filled by debris and stones up to shallow depths, which was most probably an open access to the tunnels (feature S, Figure 24).

The map in Figure 25 combines the results presented so far, based on magnetic field modelling and the integration of radar and resistivity data. It shows a comprehensive magnetization model of the main archaeological features around the Plutonium-Inferi complex, which extends the distribution of magnetization obtained by forward modelling of magnetic anomalies (Figure 17) through the inclusion of supplementary blocks. The location and shape of the additional features was determined by conversion of GPR reflection anomalies detected on amplitude maps (Figure 9) into magnetized blocks. The magnetic parameters of these blocks were inferred on the basis of the observed magnetization parameters of the soil and tuff layers and, in the case of walls and roads, taking into account of their building materials (Table 4).

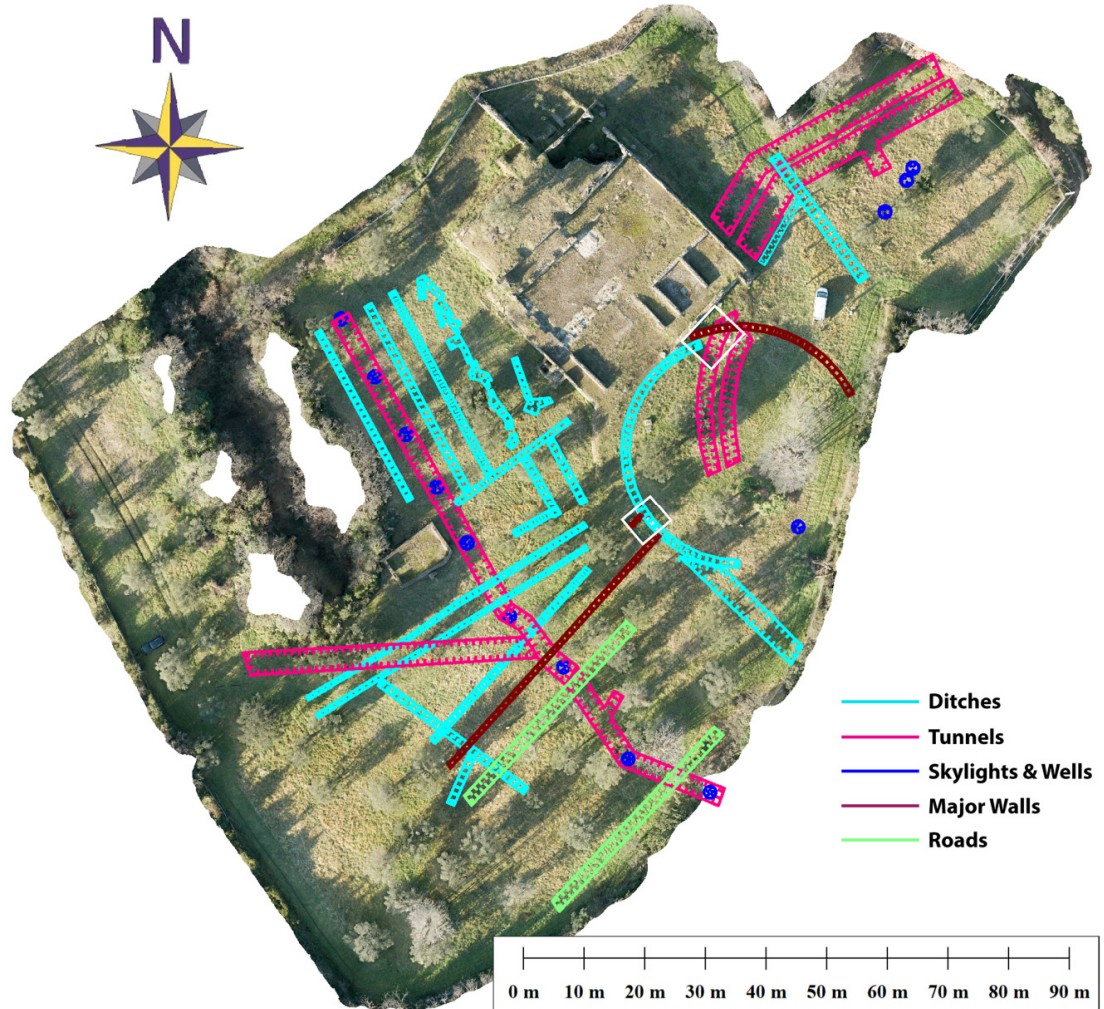

**Figure 25.** Final integrated model showing the major archaeological features in the Plutonium-Inferi area. Ditches, roads, and major wall structures are also displayed. Excavated areas are bounded by white boxes [16].

**Table 4.** Magnetization parameters of the model in Figure 25.

| Feature | D [°deg] | I [°deg] | M [A m⁻¹] | Δχ [×10⁻⁶] SI |
|---|---|---|---|---|
| Tuff layer | 4.1 | 72.8 | 4.82 | 0 |
| Soil layer | 0 | 0 | 0 | 0 |
| Air–filled cavities | 4.1 | 72.8 | −4.82 | −18127 |
| Soil–filled cavities | 4.1 | 72.8 | −4.82 | −8627 |
| Walls | 0 | 0 | 0 | −9500 |
| Roads | 0 | 0 | 0 | +5000 |

$D, I, M$ = Declination, inclination, and intensity of NRM. $\Delta\chi$ = Susceptibility contrast relative to background.

The map in Figure 25 shows a portion of the already mentioned large ditch having circular shape and a 38 m diameter, excavated at the soil–tuff interface. There is archaeological evidence that this structure was connected to the use of water, as is confirmed by the finding of lead pipes during the recent excavations [16]. The presence of this large structure, possibly equipped with a pool in the centre, suggests that the Plutonium had an impressive and monumental entrance, characterized by a circular or semi-circular water feature. Water was also channeled through pipes and ditches in order to sustain the surrounding garden [16]. Finally, the couple of tunnels detected in the northeastern sector (Area 10) seem to continue beneath the Plutonium area, where a large magnetic anomaly having a curved shape can be modelled by a pair of narrow tunnels running parallel each other at a depth of ~2.6 m (Figure 17). At the moment, there is no radar evidence of these features, but a direct probing performed by the Sotterranei di Roma has confirmed the presence of a soil-filled cavity up to ~3 m depth (M. Placidi, pers. comm).

The results presented above allow us to reconstruct the Plutonium-Inferi complex as an important landscaped area of the villa, despite its present state of deterioration if compared with other parts of the archaeological park. The map in Figure 25 shows that the main monumental building, allegedly dedicated to the cult of Pluto, was surrounded by a complex network of waterways that served a large garden and several pools. The human modifications to the landscape in this area also included large interventions such as the digging of the Inferi, a 140 m long, 20 m large, and 6 m high ditch, the excavation of a complex system of tunnels for the transport of water or supplies [1], and, possibly, earthworks for the construction of a bench terrace in the southernmost part of this area. Our work will contribute substantially to future investigations of the man-made modifications to the natural environment that took place in the making of Hadrian's Villa, where the sources attest that the Emperor attempted to reproduce various landscapes of the ancient world, as follows:

> "*Tiburtinam villam mire aedificavit, ita ut in ea et provinciarum et locorum celeberrima nomina inscriberet, velut Lycium, Academian, Prytanium, Canopum, Poecilen, Tempe vocaret. Et, ut nihil praetermitteret, etiam inferos finxit.*" (from Historia Augusta, 26, 5).

The English translation of this text reads:

> "*His villa at Tibur was marvelously constructed, and he actually gave to parts of it the names of provinces and places of the greatest renown, calling them, for instance, Lyceum, Academia, Prytaneum, Canopus, Poecile, and Tempe. And in order not to omit anything, he even made a Hades.*"

## 5. Conclusions

In this paper, we have presented a model of the tunnels running beneath the Plutonium-Inferi area and a description of buried structures that were presumably part of a garden. The latter features consist of a system of ditches that were carved on the top of the tuff. The proposed pattern of buried structures is based on the forward modelling of magnetic anomalies, supported by GPR amplitude slices and the analysis of individual GPR and ERT profiles. In addition to the primary goal of delineating the precise location of known tunnels across the Plutonium-Inferi area, we found evidence of the following: (a) Two entrances to the underground tunnels; (b) a pair of previously unknown parallel tunnels in

the eastern sector, which probably represent a way out from the villa towards the Via Tiburtina, (c) a previously unknown transversal tunnel in the western sector, probably an aqueduct of Republican age, and (d) two ancient roads directed to the Plutonium in the southern sector.

**Supplementary Materials:** The following are available online at https://zenodo.org/record/3351757#.XVIUHdIRWUl, Figure S1: Local reference frames used for the acquisition of GPR data, Figure S2: UAV orthophoto of the study area (Plutonium−Inferi complex) with indication of the excavated areas, Tables S1–S13: Relevant migrated and unmigrated GPR profiles for areas 1 through 13, Table S10A, transversal migrated and unmigrated GPR profiles for Areas 10.

**Author Contributions:** Conceptualization of the methodology, design of the field experiments, design of the modelling software, formal analysis of magnetic and GPR data, and integration of the various sources of data, A.G. and A.S.; interpretation of radar profiles, A.G. and L.C.; acquisition, treatment, and formal analysis of paleomagnetic data, L.V. and A.G.; acquisition and formal analysis of electric resistivity data, L.T. and P.B.; mineralogical analysis of rock samples, E.S.; acquisition and formal analysis of aerial photogrammetry data, P.B.; GPS data acquisition and design of the field survey geometry, A.S., P.P.P., and A.G.; geological data acquisition and analysis, P.P.P. and E.S.; magnetic and GPR data acquisition, A.G., A.S., P.P.P., L.T., and E.S.; original draft preparation, A.G.; review and editing, A.S., M.M., L.T., E.S., L.V., and L.C.; project supervision, A.S.; project coordination, M.E.G.

**Funding:** This research was funded by the Università degli Studi di Camerino, grants FAR Schettino 2016–2018 and FAR Pierantoni 2016–2018, and by the University of Oxford, Eugene Ludwig Fund, New College.

**Acknowledgments:** The authors are grateful to the Director of the Villa Adriana and Villa d'Este, Andrea Bruciati, for kindly allowing us to survey the archaeological area and to Benedetta Adembri for facilitating the research on site. We are also grateful to Francesco Ferruti and the students that helped us in the data acquisition. Finally, we thank Alessandro Bertani for his help in the acquisition and formal analysis of aerial photogrammetry data. This paper also benefited from four accurate reviews that allowed us to improve the manuscript.

**Conflicts of Interest:** The authors declare no conflict of interest. The funders had no role in the design of the study; in the collection, analyses, or interpretation of data; in the writing of the manuscript, or in the decision to publish the results.

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
