# Peer review of "Reconstruction of a Segment of the UNESCO World Heritage Hadrian’s Villa Tunnel Network by Integrated GPR, Magnetic–Paleomagnetic, and Electric Resistivity Prospections"

_remotesensing, doi:10.3390/rs11151739_

Round 1

Reviewer 1 Report

General Comments

The investigated site is famous and the authors have made a great works applying different geophysical methods, supporting the results with laboratory measurements fort analyse the magnetic response of the materials. However, the structure of the article requires being totally re-organized. The presentation of the results is confused and their discussion is not adequately supported. Despite some results are interesting their reading is hard and, for this reason, in the actual form the paper is far from the publication. I hope that the authors enhance the quality of the work and  resubmit  a new version of the paper

Specific comments

Abstract

Some information are redundant and not useful to present the work. This section should be rewritten.

The first sentence is out of the context. Add other information if you consider this information important for the paper (Raw 22)

Raws 31-36 describe confusedly the target of the paper. Reorganize the abstract considering the suggestion of the journal

Introduction

In this section it could be useful create a distinction between the archaeological context of the work and the importance of application of geophysical methodologies for archaeological issues. The present form is confused and  difficult for readers to understand.  A new subsection could be inserted for discuss more adequately the geological conditions of the site. Further the figures should be reconsidered. In particular the north is unreadable for figure 3 and in figure 2 the metric scale or a reference is missed. Figure 1 has a low resolution (change it). Finally in figure 3 there are litters and numbers that are not discussed in the paper (if are not useful delete them). The references presented are too few and not consider the geophysical problems for archaeological issues.

Methods  

Also in this section I suggest to create two subsection, in the first you can present the acquisition phase and in the second the processing of the data. Raw 163 describe a bidirectional mode of acquisition but figure 6 shows only a direction of acquisition. Where is the truth? Did you mean acquisition based on the snake mode? Check the sentence also for magnetometric acquisition. As regarding processing phase all the steps indicated between the raws 174 and 180 should be analysed and accurately described. Further, in my opinion, the strong heterogeneity of the dielectric permittivity is little realistic. Justify why areas close show values of permittivity of 20, 10,20 as in the case of the areas 1-2-3.  The metric scale and north indicator are not easily readable. Finally for magnetic acquisitions, did you use GPR or markers? Specify in the text.

The section 2.3 contains also the results. Why, if we are in the section methods? Reorganize the entire section in a more rational way.

As regarding ERT, justify the use of the two different arrays.

The aerial Photogrammetry is used for creating a DEM but its use was neglected for processing the geophysical data. So, the authors should be justify the reason to create a section for the photogrammetry when its use if not consistent for the importance of the paper.

Results

Also in this case the creation of distinct subsections could be provide a better comprehension of the text. I suggest The interpretation of the results can be presented in the paragraph of the discussion. When you submit the new version consider that the raws from 343 to 351 and 365 to 377 should be inserted in a methodological section not into the results. Further, GPR depth slices show results not clear and it is better support the interpretation of the archaeological remains with 2D radargrams. The interpretation of the resistive layer showed by the ERT (figure 16) is little realistic. You are considering what happens to the edge of the tomography where the resolution is not adequate. The radargram plotted in figure has two reference scales for Y-axis (check). The font of the A trace caption is too little. For the figures 17-18-19-20 the interpretation is not supported by the results showed that are, also, presented in a really confused way. Reorganize all the figures and resubmit a new clearer version.

Reviewer 2 Report

Paper focused on a Magnetic, paleomagnetic, radar, and electric resistivity surveys performed in the Plutonium–Inferi sector to detect buried buildings and outline a segment of the underground system of tunnels that link different zones of the villa.

This paper constitutes an interesting but standard application of the combined geophysical techniques. Results are interesting but does not lead in a scientific advanced in the archaeological geophysical field.

Improvements are request

1)     Abstract needs to be more concise, focus on location, need for investigation methods used expected outcomes.

2)     Introduction: authors should be clear how your work could be advances the state of the art or the knowledge in this field. I think that the Authors could be focused this paragraph to specify the peculiar aspects of novelty of this paper with respect to other studies performed from other authors.

3)     I believe that would be much more informative if Authors provide additional details about the survey strategy and the processing procedure.

Suggest following break down:

a) raw data initial interpretation

b) Modelling description and outcomes

c) processing steps used and why?

 d) final interpretation of processed results

4)     Please would be interesting if Author underline the depth and not the sample number in the 2D radar section.

5)     Figure 16 is unclear because I expect in the presence of voids an high resistivity values and maybe an inversion of polarity from em reflected waves. Please explain this.    

Reviewer 3 Report

Dear Authors,

congratulations for the very interesting and complete geophysical investigation you present in the article. I would like just to comment on the appllication of the electrical resistivity tomography. First, I would like to point out the fact that you chose to use the Wenner array instead of Dipole Dipole. I cannot understand that since the nature of your target requires an array sensitive to lateral resistivity changes such as the tunnels. In Figure 16, the depth of the tomography is very little according to the length of the line (31 meters). At the most part of the line is almost 3 meters and only in the cetral area of 4 meters gets the maximum depth of 5 meters, which still is too little. I beleive that the interpretation of the resistive part of the tomography (5 to 14m) as a void is excessive since (a) the resistivity value is very low, and (b) the area is very wide. 

Similalry, the evidence of the presence of a tunnel presented in Figure 19 is not clear to me since it is detected at the bottom of the resistivity section and it could be an edge effect.

In Figure 20, I understand that the two resistive areas (centers at 13 and 21m) correspond to radar hyperbolas (2 and 9m) showing the continuation of the tunnels.  In such case, how can you explain that the tunnel is not detected in the parallel radar line where only the 'S' target -interpreted as skylight- is detected?

Also, it is not clear to me if at the ERT at the bottom left of Fig. 20, the resistive areas are interpreted as tunnels. If so, I think that it is again an excessive interpretation since it is more possible that this is a geologivcal layer. In any case, inversion must be checked again since the depth of 5 meters is very small for an array of 31 meters

In the future, I would suggest to use more than one electrode arrays to be able to choose the most appropriate one according the the real earth model.

The article is very well written, just two small notices:

Line 384: ensembles (instead of ensambles)

Line 467: area (instead of region)

Reviewer 4 Report

This paper presents an interesting combination of geophysical methods to detect tunnel network in an archaeological site. I read the article with great interest. The combination of complementary NDT techniques can be considered a high interest issue as the approach presented herein can be generalized for other purposed of the Ground Penetrating Radar application.

That being stated, the manuscript is worthy to be published in Remote Sensing.

My main criticisms are:

- Section 1 "Introduction". There is not a state of the art of the techniques on the field of application of the study. Authors should include previous works and experiences, for all the geophysical techniques employed, in the detection of buried buildings and tunnelling. Why authors have chosen these techniques?

- In page 5, line 147: Authors should explain how they obtained a maximum depth of penetration of 2.5 m? If they use particular values of attenuation and/or velocity, or bibliography. Please, state in the manuscript. 

- In page 5, lines 158-172: Please, provided the model of the GPS system used and its accuracy. 

- In page 5, line 172: Please, provide the software used for GPR data processing. Is it commercial software or some own software? 

- In page 7, lines 203-208: Have you considered some threshold for amplitude slices? Please, include some short explanation for the processes of normalization, knitting and equalization. 

- In page 8, lines 221-223.  Please, provide the software used for data processing. Is it commercial software or some own software? 

- In page 10, section 2.5 "Aerial photogrammetry". Please, provided the model of the RGB camera used and focal length.

- In page 10, line 304: How many control points? 

- Please, provide also the conditions of the photos acquisition: time of the day, sunny/cloudy, orientation, etc. The orthoimage (Figure 10) obtained could be improved avoiding the shadows. 

- Section 3 "Results". Authors should include the results of the electrical resistivity method, as well as to discuss the resistivity values obtained in correspondance with the features of interest. Currently, the ERT values are also included in the discussion section, but there is no any quantitative interpretation. 

- Section 5 "Conclusions". This section should be improved. Authors should include a discussion of the different methods used: advantages or disadvantages with the objectives of your study, limitations and, most important, the interest/benefits of the combination. 

Round 2

Reviewer 1 Report

As regarding the title, it should be better to change the title in comparison of geophysical and aerial analysis because the integration of the data is poor and also in a few cases is tried.  So it is hard to affirm that the paper presents an integration of data.

Paragraph 1 “Introduction” should be changed in the archaeological introduction.

The sub-paragraph 1.3 is interesting but should be inserted in a paragraph that describes the methodology adopted and for this reason, it should be rearranged and deeply changed. Indeed, this is particularly true for the raws 149 to 197 where the methodology adopted is presented. The section method of the second paragraph should be to collect this discussion.

Totally absent references about the use of geophysical methods in archaeological contexts.

As regards paragraph 2.1 it is really strange the orientation of the GPR profiles, for some areas the profiles were acquired according to the direction north-south; for other east-west. Did you consider the polarity and directivity of GPR method? Why did not you acquire in two perpendicular directions in each considered areas? Table 1 should be contain the references of each software and avoid to insert the owners of the software

Table 2; I have strong doubts about the permittivity of area 8 that shows significant differences with respect to the closer areas 7 and 9.

The authors have presented GPR and magnetic acquisition and processing phases separately, why don’t use the same scheme for presenting paleomagnetic, ERT and UAV data collection and elaboration?

Results are interesting but no integration is made as expected from the title of the work. I suggest inserting a new subparagraph in the results where the integration, shortly described in the paragraph discussion, is widely presented. Finally, how is it possible interpret even the types of structures having only a simple em reflection recorded by the GPR? Figure 9 is hard to accept in the present form for geophysicists.

As regarding the discussion, it is possible to identify only a comparison of GPR, ERT and magnetic data that reinforces the fact that the paper is a comparison of different data acquired with geophysical techniques with no integration. The research appears to be, despite the significant work made by the authors, more like a report.

Reviewer 2 Report

Paper was improved and merit to be published
